# Surgical and regional treatments for colorectal cancer metastases in older patients: A systematic review and meta-analysis

**Nicola de'Angelis**[1]*, **Capucine Baldini**[2], **Raffaele Brustia**[3], **Patrick Pessaux**[4], **Daniele Sommacale**[5], **Alexis Laurent**[1], **Bertrand Le Roy**[6], **Vania Tacher**[7], **Hicham Kobeiter**[7], **Alain Luciani**[7], **Elena Paillaud**[8], **Thomas Aparicio**[9], **Florence Canuï-Poitrine**[10], **Evelyne Liuu**[11]

1 Unit of Digestive, Hepato-Pancreato-Biliary Surgery, Henri Mondor Hospital, AP-HP, University of Paris Est, UPEC, Créteil, France, 2 Drug Development Department, Gustave Roussy Cancer Campus, University Paris-Saclay, Villejuif, France, 3 Department of Hepato-biliary and Liver Transplantation Surgery, Pitié-Salpêtrière University Hospital, APHP, Paris, France, 4 Digestive, and Endocrine Surgery, Nouvel Hôpital Civil, Université de Strasbourg, and U1110 Inserm, Institute of Viral and Liver Disease, Strasbourg, France, 5 Department of General and Digestive Surgery, Hôpital Robert Debré, Centre Hospitalier Universitaire de Reims, Université de Reims Champagne-Ardenne, Reims, France, 6 Department of Digestive Surgery, University Hospital of Saint-Etienne, Saint-Priest-en-Jarez, France, 7 Departement of Radiology, Henri Mondor Hospital, AP-HP, University of Paris Est, UPEC, Créteil, France, 8 Hopital Europeen Georges Pompidou, Department of Geriatrics, Paris, France, 9 Gastroenterology and Digestive Oncology Department, Saint Louis Hospital, AP-HP, and University of Paris, Paris, France, 10 Department of Epidemiology and Biostatistics, Henri Mondor Hospital, AP-HP, University of Paris Est, UPEC, Créteil, France, 11 Department of Geriatrics, CHU La Milétrie, Poitiers University Hospital, Grand Poitiers, France

* nicola.deangelis@aphp.fr

## Abstract

### Objective

The present study explored the existing literature to describe the outcomes of surgical and regional treatments for colorectal cancer metastases (mCRC) in older patients.

### Methods

A literature search was conducted in PubMed, EMBASE, Cochrane and ClinicalTrials.gov for studies published since 2000 that investigated the short- and long-term outcomes of regional treatments (surgical or non-surgical) for mCRC in patients aged ≥65 years. Pooled data analyses were conducted by calculating the risk ratio (RR), mean differences (MD) and hazard ratio (HR) between older and younger patients or between two different approaches in older patients.

### Results

After screening 266 articles, 29 were included in this review. These studies reported the outcomes of surgery (n = 19) and non-surgical local ablation treatments (n = 3) for CRC metastases in older vs. younger patients or compared the outcomes of different interventions in

**Funding:** No funding or research grant was received to perform this systematic review and meta-analysis. No external financial support was received. The study was promoted by the Societé Francophone d'Oncogeriatrie (SoFOG), but no financial contribution was received from the scientific society. The authors' affiliations (universities/hospitals) provided support exclusively in form of salaries and did not have any additional role in the study design, data collections and analyses, decision to publish, or preparation of the manuscript.

**Competing interests:** Pr P Pessaux is an orator for Integra and a co-founder of VirtualiSurg. This affiliation does not alter our adherence to PLOS ONE policies on sharing data and materials. All other authors have nothing to disclose, and all authors have declared that no competing interests exist in relation to the matter of this study.

older patients (n = 7). When comparing older vs. younger patients undergoing liver surgery for mCRC, pooled data analysis showed higher postoperative mortality [RR = 2.53 (95%CI: 2.00–3.21)] and shorter overall survival [HR = 1.17 (95%CI: 1.07–1.18)] in older patients, whereas no differences in operative outcomes, postoperative complications and disease-free survival were found. When comparing laparoscopy vs. open surgery for liver resection in older mCRC patients, laparoscopy was associated with fewer postoperative complications [RR = 0.27 (95%CI: 0.10–0.73)].

## Conclusion

Liver resection for mCRC should not be disregarded *a priori* in older patients, who show similar operative and postoperative outcomes as younger patients. However, clinicians should consider that they are at increased risk of postoperative mortality and have a worse overall survival, which may reflect comorbidities and frailty.

## Introduction

With the increase of life expectancy, the proportion of people aged 65 years and over has increased five-fold during the last 15 years [1]. Recent approaches have forecasted an increase in life expectancy by 4.4 years for both sexes by 2040, exceeding 85 years in many developed countries [2]. Indeed, although an increase in functional impairment and frailty is observed with aging, life expectancy for those aged between 80 years and 85 years is still 8 years [3–5], with a consequent greater chance for clinicians to diagnose diseases and treat patients at an advanced age.

Colorectal cancer (CRC) is the third most incident cancer in adults [6, 7] and the second most common cause of cancer-related death in Europe [8]. In 20–25% of cases, CRC presents with simultaneous liver metastases (American Joint Committee on Cancer [9], AJCC stage IV) and 85% of these lesions are not resectable at diagnosis [7, 10, 11]. Moreover, an additional 25–50% of CRC patients will develop metachronous metastases after the resection of the primary tumor, with the liver as the most frequent initial recurrence site [12–14]. In the case of metastatic CRC (mCRC), a multimodality treatment is required [15, 16]. Hepatectomy for colorectal liver metastasis (CRLM) offers the highest cure rate and is indicated for adult patients with primarily resectable disease or after downstaging chemotherapy [11, 17, 18], with a 5-year survival rate that ranges from 35% to 60% [19]. Alternative treatments include chemotherapy (CT) regimens, local ablation therapies, radio-embolization and hepatic intra-arterial chemotherapy [15, 20–22].

The yearly incidence of CRC has increased in people aged 75 years or older [23]. In France, 45% of new cases have occurred in patients aged 75 years or older [24]. In general, there is less frequency of chemotherapy or liver surgery for mCRC in older populations [25, 26]. Non-surgical therapies are favored in older patients with the assumption that advanced age and the presence of comorbidities yield a higher risk of surgical morbidity and mortality [14]. Age is often considered a risk factor for poorer postoperative outcomes; however, compelling evidence supports that it is not the actual chronological age of the patient that constitutes a risk for surgery but rather the quality of aging, comorbidity and the functional status that define the condition of frailty [27–29]. Therefore, caring for older patients with mCRC is an ongoing challenge and to date, there is still a lack of guidelines to support the decision of the optimal strategy for the management of mCRC in older patients (age ≥ 65 years)[14, 25, 30].

This study aimed to explore the current literature to *i*) describe the current trend of regional treatments for mCRC in older patients and *ii)* evaluate the clinical and oncological outcomes of surgery and regional treatment options in older vs. younger patients.

## Methods

### Study design

This is a systematic review and meta-analysis designed to describe and evaluate the outcomes of different regional treatments (i.e., surgery, radiofrequency, cryotherapy, microwave abla-tion, electroporation, and radioembolization) in the case of treatable mCRC in older patients. The present report is structured according to recommendations by the Preferred Reporting Items for Systematic Reviews and Meta-Analysis (PRISMA) guidelines [31, 32], and the study protocol has been registered in the PROSPERO database (provisional registration number: 132956).

### Eligibility criteria for study inclusion

Studies were eligible for inclusion if they met the criteria established by the following PICOS framework:

*Patients*: patients aged ≥65 years and diagnosed with mCRC (AJCC stage IV)[33, 34].

*Intervention*: regional treatments including curative-intent surgical resection, radioemboli-zation, cryotherapy, microwave ablation, electroporation, regional hepatic intra-arterial che-motherapy, and chemoembolization for synchronous or metachronous mCRC [22].

*Comparison*: according to the patients' age (e.g., surgical resection in older vs. surgical resection in younger patients) or the type of intervention (e.g., surgical resection vs. chemo-therapy in older patients).

*Outcomes*: operative and postoperative outcomes (e.g., morbidity, mortality) and survival rates (overall survival, OS; disease-free survival, DFS).

*Study design*: randomized controlled trials (RCTs), non-randomized controlled trials (NRCTs), and observational case-control and cohort studies.

### Search strategy

Relevant human studies were identified up to March 2019 from the following online available databases: MEDLINE (through PubMed), EMBASE, Cochrane Library, and ClinicalTrials.gov register. For each database, a specific research equation was formulated using the following key words and/or MeSH terms: metastatic colorectal cancer, metastasis, elderly, older, age ≥65, age ≥70, age ≥80, surgery, surgical resection, liver resection, metastasectomy, radiofre-quency ablation, radioembolization, local ablation treatment, cryotherapy, microwave abla-tion, electroporation, regional treatment. In addition, manual searches in the references sections of eligible studies and relevant review articles were crosschecked to identify additional records. The literature search was limited to the time frame January 2000 –March 2019, and only English literature was considered.

### Study selection

A parallel, double blind screening procedure of titles and abstracts was carried out by two reviewers (NdeA and EL). To enhance sensitivity, records were removed only if both reviewers excluded the record at the title/abstract level. The reviewers' consistency was assessed by the kappa test. Subsequently, both reviewers performed a full-text analysis of the articles and

proceeded to the final selection phase. Any disagreement between the two reviewers was resolved by consensus with a third reviewer (CB).

## Data extraction and quality assessment

The following variables were extracted from the selected articles and collected in an excel spreadsheet: authors, year of publication, journal, country, study time frame, study design, patient population, type of regional treatments, type of surgical procedure, intraoperative and postoperative outcomes, 90-day morbidity (overall postoperative complications; type and severity of postoperative complications according to Dindo-Clavien classification, including pulmonary complications, hepatic complications, and major complications) and 30-, 60- or 90-day mortality, survival (overall, disease-free, cancer-specific) at any reported time point (1, 2, 3, 5 years or more after the intervention).

The risk of bias was assessed using the Newcastle–Ottawa Scale (NOS) for case-control and cohort studies [35]. Both data extraction and the quality assessment of the study were performed by the two reviewers (NdeA and EL) independently and compared to reach a consensus (if necessary) with the third reviewer (CB).

## Pooled data analyses

Data from the included studies were used for a qualitative and quantitative synthesis according to the patients' age or the type of intervention. For binary outcomes, the risk ratio (RR) and 95% confidence intervals (CI) were estimated using the Mantel-Haenszel method. For continuous data, the mean differences and 95%CI were estimated using inverse variance weighting. Outcome measures were extracted as the mean (SD) or median (interquartile range) as provided. To calculate the mean values from the median, we applied the method described by Hozo et al. [36].

To compare OS and DFS between older and younger patient groups or between different types of interventions for mCRC, we calculated the hazard ratio (HR) and 95%CI as described by Tierney et al. [37]. Heterogeneity was assessed by the $I^2$ statistic, and values of 25%, 50%, and 75% were considered low, moderate, and high heterogeneity, respectively [38, 39]. Random effect model was used as considered a more precise estimator when there is in between study heterogeneity in true effects. Whenever indicated, sensitivity analyses were performed to test degree of certainty of the results. A $p < 0.05$ identifies significant pooled effects. Meta-analyses were performed using the Cochrane Collaboration software, RevMan (Version 5.3).

# Results

## Literature search and selection

Results of the literature review in the considered databases and the step-by-step study selection are shown in Fig 1. Overall, the combined search on the different databases identified 266 articles. After removing duplicates and non-pertinent studies upon title and abstract, 146 articles underwent a full-text evaluation. Of these, 124 were excluded because they were not pertinent to the review question. Finally, 29 articles were selected and eligible for pooled data analyses (availability of outcome data). The two reviewers had an optimal interexaminer agreement in the selection process (kappa: 0.978).

## Study characteristics

Studies were published predominantly after 2013 (62% of the studies), whereas no study was published before 2004 (S1 Fig). These studies were conducted in Europe (n = 15), North

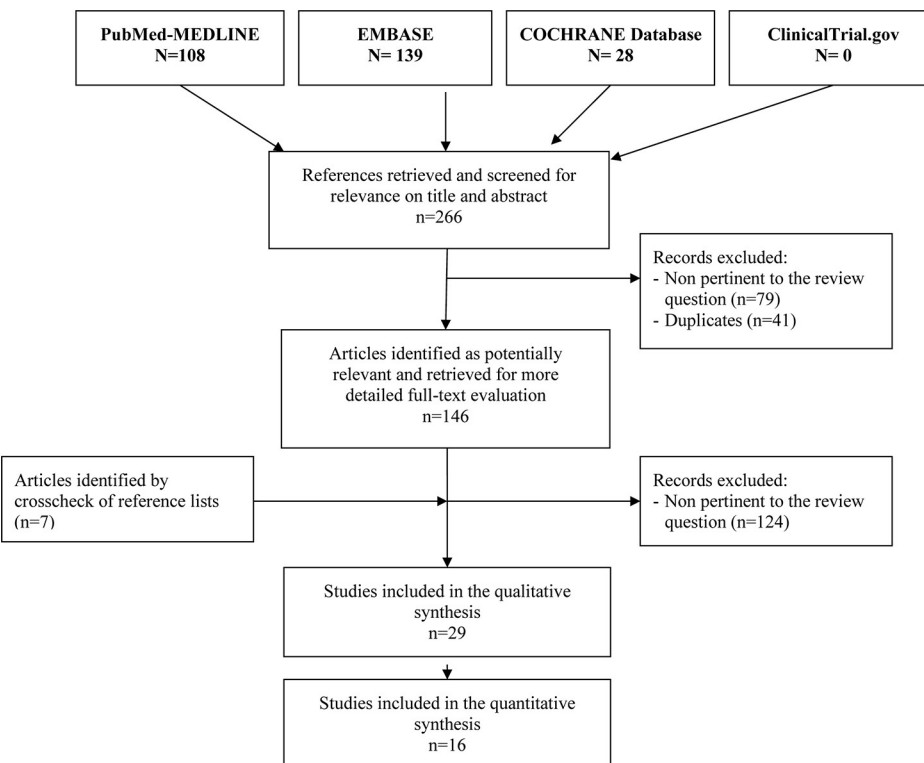

PubMed equation: (colon[Title] OR colorectal[Title]) AND (cancer*[Title] OR neoplasm*[Title]) AND ("radiofrequency"[All Fields] OR "surgery"[All Fields] OR "radioembolization"[All Fields] OR OR "cryotherapy"[All Fields] OR "regional treatments"[All Fields] OR "electroporation"[All Fields] OR "intraarterial"[All Fields] OR "hepatectomy"[All Fields] OR "liver resection"[All Fields] OR "metastasectomy"[All Fields]) AND metastatic*[Title] AND ( old*[Title/Abstract] OR elderly[Title/Abstract] OR "aged 65" [Title/Abstract] OR "more than 65" [Title/Abstract] OR "older than 65" [Title/Abstract] OR "aged≥65" [Title/Abstract] OR "aged ≥65" [Title/Abstract] OR "aged≥ 65" [Title/Abstract] OR "aged ≥ 65" [Title/Abstract] OR "aged>65" [Title/Abstract] OR "aged >65" [Title/Abstract] OR "aged> 65" [Title/Abstract] OR "aged > 65" [Title/Abstract] OR "age≥65" [Title/Abstract] OR "age ≥65" [Title/Abstract] OR "age≥ 65" [Title/Abstract] OR "age ≥ 65" [Title/Abstract] OR "age>65" [Title/Abstract] OR "age >65" [Title/Abstract] OR "age> 65" [Title/Abstract] OR "age > 65" [Title/Abstract] OR "aged 70" [Title/Abstract] OR "more than 70" [Title/Abstract] OR "older than 70" [Title/Abstract] OR "aged ≥70" [Title/Abstract] OR "aged≥ 70" [Title/Abstract] OR "aged ≥ 70" [Title/Abstract] OR "aged>70" [Title/Abstract] OR "aged >70" [Title/Abstract] OR "aged> 70" [Title/Abstract] OR "aged > 70" [Title/Abstract] OR "age≥70" [Title/Abstract] OR "age ≥70" [Title/Abstract] OR "age≥ 70" [Title/Abstract] OR "age ≥ 70" [Title/Abstract] OR "age>70" [Title/Abstract] OR "age >70" [Title/Abstract] OR "age> 70" [Title/Abstract] OR "age > 70" [Title/Abstract] OR "aged 75" [Title/Abstract] OR "more than 75" [Title/Abstract] OR "older than 75" [Title/Abstract] OR "aged≥75" [Title/Abstract] OR "aged ≥75" [Title/Abstract] OR "aged≥ 75" [Title/Abstract] OR "aged ≥ 75" [Title/Abstract] OR "aged>75" [Title/Abstract] OR "aged >75" [Title/Abstract] OR "aged> 75" [Title/Abstract] OR "aged > 75" [Title/Abstract] OR "age≥75" [Title/Abstract] OR "age ≥75" [Title/Abstract] OR "age≥ 75" [Title/Abstract] OR "age ≥ 75" [Title/Abstract] OR "age>75" [Title/Abstract] OR "age >75" [Title/Abstract] OR "age> 75" [Title/Abstract] OR "age > 75" [Title/Abstract] OR "aged 80" [Title/Abstract] OR "more than 80" [Title/Abstract] OR "older than 80" [Title/Abstract] OR "aged≥80" [Title/Abstract] OR "aged ≥80" [Title/Abstract] OR "aged≥ 80" [Title/Abstract] OR "aged ≥ 80" [Title/Abstract] OR "aged>80" [Title/Abstract] OR "aged >80" [Title/Abstract] OR "aged> 80" [Title/Abstract] OR "aged > 80" [Title/Abstract] OR "age≥80" [Title/Abstract] OR "age ≥80" [Title/Abstract] OR "age≥ 80" [Title/Abstract] OR "age ≥ 80" [Title/Abstract] OR "age>80" [Title/Abstract] OR "age >80" [Title/Abstract] OR "age> 80" [Title/Abstract] OR "age > 80" [Title/Abstract]) AND (English[lang] OR French[lang]) AND ((Clinical Conference[ptyp] OR Clinical Study[ptyp] OR Clinical Trial[ptyp] OR Clinical Trial, Phase II[ptyp] OR Clinical Trial, Phase III[ptyp] OR Clinical Trial, Phase IV[ptyp] OR Comparative Study[ptyp] OR Letter[ptyp] OR Meta-Analysis[ptyp] OR Multicenter Study[ptyp] OR Observational Study[ptyp] OR Pragmatic Clinical Trial[ptyp] OR Randomized Controlled Trial[ptyp] OR Research Support, American Recovery and Reinvestment Act[ptyp] OR Research Support, N I H, Extramural[ptyp] OR Research Support, N I H, Intramural[ptyp] OR Research Support, Non U S Gov't[ptyp] OR Research Support, U S Gov't, Non P H S[ptyp] OR Research Support, U S Gov't, P H S[ptyp] OR Review[ptyp] OR Research Support, U.S. Government[ptyp] OR systematic[sb] OR Classical Article[ptyp] OR Congresses[ptyp] OR Consensus Development Conference[ptyp] OR Consensus Development Conference, NIH[ptyp] OR Controlled Clinical Trial[ptyp] OR Corrected and Republished Article[ptyp] OR English Abstract[ptyp] OR Evaluation Studies[ptyp] OR Guideline[ptyp] OR Journal Article[ptyp] OR Practice Guideline[ptyp] OR Published Erratum[ptyp] OR Retracted Publication[ptyp] OR Retraction of Publication[ptyp] OR Scientific Integrity Review[ptyp] OR Validation Studies[ptyp])

**Fig 1. Flowchart of literature search and study selection process.**

America (n = 7), Asia and the Pacific (n = 7). There was no RCT. All of the studies were retrospective, of which 7 were population-based cohort studies [25, 26, 40–44], 16 retrospective analyses of patient cohorts [45–60], 4 case-control studies [61–64], and 2 propensity score matched (PSM) case-control studies [65, 66]. Overall, 5 studies (17.8%) were multicentric [48, 49, 51, 59, 66]. The study time frames spanned from 1986 to 2017, with a mean of 9.64 years (range: 8 months-22 years).

Twenty-two studies compared the outcomes of local ablation therapies in older patients vs. younger patients. Of these, 19 studies reported the outcomes of surgical resection for CRC metastases (Table 1). Specifically, 16 studies focused on liver resection for CRLM [40, 45, 47, 50–52, 54–57, 59–64], and 3 studies investigated the outcomes of multimodality treatments, including surgical resection for CRC metastases [25, 42, 44]. Only 3 studies reported the outcomes of non-surgical regional treatments (including radioembolization [RE], radiofrequency ablation [RFA], and high-dose-rate brachytherapy [HDR-BT]) for mCRC in older vs. younger patients (Table 2)[49, 53, 58]. The remaining 7 studies compared the outcomes of different types of regional interventions in older patients [26, 41, 43, 46, 48, 65, 66](Table 3).

## Definition of older patients

The definition of older patients largely varied among the studies. The majority of the studies (55.1%) used an age cut-off of ≥70 years [46, 47, 49, 50, 52–59, 62–64, 66] to define older people. Two studies defined older patients aged ≥65 years[41, 65] and 6 patients aged ≥75 years [25, 26, 40, 43, 45, 48, 51, 60, 61]. One study defined older patients as those aged ≥80 years[42] or ≥85 years [44] and compared them to younger patients. Few studies compared more than 2 age groups, with a category of very old patients, whose age cut-off varied from 75 to 85 years old [44, 51, 61]. For the sake of clarity, the terminology has been unified as older vs. younger patients in the present systematic review, although in each study different terms were used (e.g., old, older, elderly, non-elderly, young, younger patients).

Scores of general status or comorbidity were reported in only 48.2% of the studies. Eight studies used the Charlson Comorbidity Index (CCI)[25, 40, 41, 43, 52, 53, 62, 64], 5 studies used the Eastern Cooperative Oncology Group Performance Status (ECOG-PS) [44, 48, 49, 58, 63], and one study used the Prognostic Nutritional Index (PNI)[57]. None of the studies reported frailty status or morbidity, mood, cognitive function or social environment parameters in older patients.

## Outcomes of surgical resection of mCRC in older vs. younger patients

The 19 studies that reported on the outcomes of surgical resection for CRC metastases presented different study designs, populations, and settings (Table 1). Overall, the outcomes of liver resection for CRLM were investigated in 7579 older patients, 179 very old patients, and 15904 younger patients [25, 40, 42, 44, 45, 47, 50–52, 54–57, 59–64, 67]. The use of portal vein embolization before surgery was reported in 5 studies [45, 54, 61, 63, 64], which was not applied differently between older and younger patients.

Although the disease stage was not systematically reported, the metastases characteristics were described in the majority of the studies (84.2%), with seldom differences between the groups. However, older patients usually presented with comorbidities, greater ASA score, and lower treatment rate than younger patients. In general, older patients were less likely to undergo major liver resection and less likely to receive perioperative chemotherapy.

Ten studies found similar postoperative complication rates between older and younger patients [47, 48, 51, 52, 56, 57, 61–64], whereas 5 studies reported a higher incidence of postoperative complications in the older [50, 54, 55, 59, 60]. Four studies did not report morbidity

Table 1. Summary of study characteristics and study outcomes of articles comparing different age groups of patients (older vs. younger patients) receiving surgery for metastatic colorectal cancer.

| Ref. | Study design and time frame | Type of intervention | Number of patients | Metastasis characteristics | Neoadjuvant chemotherapy | Morbidity rate | Mortality rate | R0 resection | OS | DFS | Predictors of morbidity and survival |
|---|---|---|---|---|---|---|---|---|---|---|---|
| **Nagano et al. 2005** | Retrospective cohort study 1992–2004 | **Liver resection** | 212 Older (≥70 yo) = 62 Younger (<70 yo) = 150 | **Single nodule:** 34/61 Older patients vs. 93/150 Younger **Metachronous lesion:** 35/61 Older patients vs. 74/150 Younger (NSD) **Maximum tumor size:** 43.8 mm for Older vs. 35.8 mm for Younger (p = 0.071) | Neoadjuvant hepatic arterial infusion Older: 11.3% Younger: 16% (p = 0.377) | Older: 19.7% Younger: 23.3% (p = 0.562) | Older: 0% Younger: 0.49% | Older: 48.1% Younger: 52.2% (NSD) | 1-, 3-, 5-year OS Older: 79.4%, 46.5%, 34.1% Younger: 90.6%, 62.8%, 53.1% (p = 0.01) | 5-year DFS: Older: 50.7% Younger : 46.5% (NSD) | • The rate of non treatment for hepatic recurrence was **higher in Older patients** than that in younger patients (29.2% versus 10.6%; p = 0.04) • **Advanced chronologic age cannot be regarded as a medical contraindication to hepatic resection for CRLM in patients ≥70 years.** |
| **Figueras et al. 2007** | Retrospective cohort study 1990–2006 | **Liver resection** | 648 Older (≥70 yo) = 160 Younger (<70 yo) = 488 | **Synchronous metastases:** 39% of Older vs. 49% of Younger (p = 0.036) **Isolated CRLM:** 51% of Older vs. 42% of Younger (p = 0.03). **Size of the CRLM:** 4.2±2.6 cm for Older vs. 3.7±2.1 cm for Younger (p = 0.009). **Lesions larger than 10 cm:** 5% of Older vs. 1% of Younger | Older: 20% Younger: 27% (p = 0.06) | Older: 41% Younger: 34% (p = 0.008) | Older: 8% Younger: 3% (p = 0.008) | Older: 85% | 1-, 3-, 5-year OS: Older: 82%, 48%, 36% Younger: 88%, 62%, 45% (p = 0.0069) | 1-, 3-, 5-year DFS: Older: 68%, 34%, 30% for Younger: 68%, 32%, 25% (p = 0.71) | • Only tumoral size > 10 cm significantly increased the postoperative mortality risk in the Older group. • Only 50% of the Older patients received adjuvant chemotherapy compared to 70% in the Younger group |
| **Mazzoni et al. 2007** | Retrospective single center cohort study 1987–2002 | **Liver resection** Including: wedge resection, segmentectomy, right and left lobectomy | 197 Older (≥70 yo) = 53 Younger (<70 yo) = 144 | Mean lesion size 2.8 cm in both groups. LM were multiple in 103 cases and limited to one lobe in 151 patients. All patients but 84 were treated for hepatic metachronous metastases. | Not reported | Older: 20.7% Younger: 14.6% (p = 0.18) | Older: 5.7% Younger: 2.1% (p = 0.19) | Older: 83% Younger: 86.8% (NSD) | Median OS Older : 28 months Younger: 31 months (p = 0.30) | Not reported | • The number of Clinical Risk Score parameters and the microscopical involvement of the hepatic resectional margin were found to directly affect survival • Age by itself may not be a contraindication to surgery. |
| **Mann et al. 2008** | Retrospective cohort study 1999–2005 | **Liver resection** Including anatomical resections, extended procedures and extra-anatomical resections performed with curative intent | 191 Older (≥70 yo) = 49 Younger (<70 yo) = 142 | Not reported | Not reported | Older: 30.6% Younger: 19% | At 30 days : Older: 0% Younger: 2% At 60 days: Older: 4% Younger: 3% (NSD) | Not reported | 1-, 3-, 5-year OS: Older: 89%, 38%, 31% Younger:. 88%, 54%, 43% (NSD) | 1-, 3-, 5-year DFS: Older: 76%, 35%, 29%, Younger: 62%, 38%, 32%, | • Aggressive surgical policy towards CRLM in Older patients is associated with low peri-operative morbidity and mortality, as well as good long-term outcomes, thus justifying its use. |

(Continued)

Table 1. (Continued)

| Ref. | Study design and time frame | Type of intervention | Number of patients | Metastasis characteristics | Neoadjuvant chemotherapy | Morbidity rate | Mortality rate | R0 resection | OS | DFS | Predictors of morbidity and survival |
|---|---|---|---|---|---|---|---|---|---|---|---|
| **Adam et al. 2010** | Retrospective multicenter cohort study 1986–2008 | **Liver resection** Including major and minor hepatectomies | 7764 Older (≥70 yo) = 1624 Younger (<70 yo) = 6140 | **Synchronous CRLM:** 661 Older vs. 2924 Younger **Metachronous CRLM:** 913 Older vs. 2821 Younger **Maximum diameter > 50 mm:** 372 Older vs. 1302 Younger **CRLM >3:** 162 Older vs. 1206 Younger **Bilateral CRLM:** 405 Older vs. 2224 Younger | Older: 33.9% Younger: 33.2% | Older: 32.3% Younger: 28.7% (p<0.001) After major hepatectomy: Older: 37.8% Younger : 35.2% (p = 0.19) | At 60 days: Older: 3.8% Younger: 1.6% (p<0.001) After major hepatectomy: Older: 5% YOunger: 2.2% (p<0.001) | Not reported | 3-year OS: Older: 57.1% Younger: 60.2% (p< 0.001) | 3-year DFS: Older: 37% Younger: 31.9% (p = 0.051) Recurrence rate: Older: 28.1% Younger: 35.6% (p>0.001) | • Within the Older group, preoperative CT was a risk factor for postoperative morbidity • Within the Older group: >3 CRLM at diagnosis (RR = 1.63 (95%CI: 1.13-2.36), bilateral CRLM (RR = 1.39 (1.04–1.87), and concomitant extrahepatic disease (RR = 1.56 (1.08–2.23)) were predictors of mortality at 60 days |
| **Di Benedetto et al. 2011** | Retrospective matched cohort study 2002–2009 | **Liver resection** Associated surgery (major operations performed during the liver resection) for 15.6% Older vs. 28.1% for Younger (p = 0.05) | 64 Older (≥70 yo) = 32 Younger (<70 yo) = 32 | **Synchronous CRLM:** 22 Older vs. 23 Younger | Administered for unresectable liver metastases Older: 62.5% Younger: 81.3% | Older: 28.1% Younger: 34.4% (p>0.99) | At 30 days: 0% in both groups At 60 days: Older: 3% Younger: 0% (NSD) | 75% in both groups | 1-, 3-, and 5-year OS: Older: 84.1%, 51.9%, 33.3% Younger: 93.6%, 63%, 28%, (p = 0.50) | 1-, 3-, and 5-year DFS: Older: 67.9%, 29.2%, 19.5% Younger: 57.6%, 32.9%, 16.4%, (p = 0.72) | Not reported |
| **Cannon et al. 2011** | Retrospective case-control study Not reported | **Liver resection** Including synchronous resection of colon and liver; major hepatectomy (52% Older vs. 53% Younger; p = 0.9). Laparoscopy in 12.5% of Older vs. 19.8% of Younger (p = 0.3) | 279 Older (≥70 yo) = 59 Younger (<70 y.o.) = 220 | **Number of CRLM:** 2.02 for Older vs. 2.61 for Yonger (p = 0.4) **Mean size:** 4.39 cm for Older vs. 4.61 cm for Younger (p = 0.1) | Overall 58% of patients (NSD between the groups) | Older: 52.5% Younger: 48.2% (p = 0.5) | At 90 days: Older: 0% Younger: 4.1% (p = 0.2) | Older: 81.4% Younger: 88.6% (p = 0.14) | 1-, 3-, 5-year OS: Older: 91.2%, 47.6%, 20.9% Younger: 92.3%, 59.6%, 35.1% (p = 0.07) | 1-, 3-, 5-year DFS: Older: 78.3%, 22.3%, 16.7% for Younger : 74.4%, 37.9%, 18.9% (p = 0.15) | • Need of transfusion was independent predictors of postoperative morbidity (OR for non transfused patients = 0.45 [95%CI: 0.25–0.79]) • Age was not associated with shortened OS and DFS. • Fong score and BMI <20 were independent predictors of OS • Fong score was predictor of DFS |
| **Kulik et al. 2011** | Retrospective cohort study 1994–2008 | **Liver resection** Including 51.8% minor and 48.2% major resections | 939 Older (≥70 yo) = 190 Younger (40–69 y.o.) = 719 Young (<40) = 29 | **Synchronous CRLM:** 334 patients **Unilobar CRLM:** 654 patients **One CRLM:** 491 patients **Mean size:** 56.78 mm (range: 3–315 mm; SD: 39.0) | Overall: 53.2% | Older: 12.4% Younger: 15.3% (p = 0.24) | Older: 0.54% Younger: 1.26% (NSD) | Older: 96.7% Younger: 96.6% (p = 0.663) | 5-year OS: Older: 31.2% Younger: 37.5% Young: 21.6% | Not reported | • For the whole sample, metastases diameter >50 mm, rising number of transfusions (>6) needed during surgery, duration of surgical procedure >210 min, and age ≥70 years were predictors of poorer OS. • For Older patients, the rising number of transfusions (>6) needed during surgery was an independent predictor of OS (HR: 3.64 (95%CI: 1.31–10.11) |

(Continued)

Table 1. (Continued)

| Ref. | Study design and time frame | Type of intervention | Number of patients | Metastasis characteristics | Neoadjuvant chemotherapy | Morbidity rate | Mortality rate | R0 resection | OS | DFS | Predictors of morbidity and survival |
|---|---|---|---|---|---|---|---|---|---|---|---|
| **Cook et al. 2012** | Retrospective cohort study 1989–2009 | **Liver resection** All elective open hepatic resections | 1279 Older (≥75 yo) = 151 Younger (<75 yo) = 1292 | **Synchronous metastases:** 37.1% for Older vs. 39.6% for Younger **Unilateral CRLM:** 76.2% for Older vs. 72.4% for Younger ≥ **liver metastases:** 54.3% for Older vs. 69.8% for Younger | Older: 43% Younger: 55.7% (p = 0.003) | Older : 32.5% Younger : 21.2% (p = 0.02) | Older: 7.3% Younger: 1.3% (p = 0.001) | Not reported | Median OS: Older : 44.1 (range 38.4–56.8) months Younger : 43.6 (range 40.2–47.0) months (p = 0.697) | Not reported | • **Not reported** |
| **Kumar et al. 2013** | Population-based retrospective cohort study 2006–2012 | **Multi treatments** Surgery (1211), hepatic metastatectomy (292), lung metastatectomy (56), CT (1291). Liver-only metastases were reported in 35.7% Older vs. 39.7% Younger | 2314 Older (≥80 yo) = 676 Younger (<80 y.o.) = 1638 | **Synchronous CRLM:** 62.3% of Older vs. 65.3% of Younger >**2 sites of metastatic disease:** 8% of Older vs. 9.2% of Younger (NSD) | CT: 28.1% of Older vs. 68.2% of Younger Targeted therapies with monoclonal antibodies: 2.4% of Older and 16% of Younger | Not reported | Not reported | Not reported | Older: 8.2 months Younger: 19.2 months (p<0.0001) | Not reported | • **Older (≥80 yo) were less likely to receive intervention for their mCRC and had poorer survival.** • The survival of selected Older patients who received CT was similar to the survival of those younger despite the receipt of single-agent therapy. |
| **Doat et al. 2014** | Retrospective population-based national cohort study April-December 2009 | Multi treatments **Surgery of the primary tumour and metastases was** significantly less frequent among Olders | 31665 Older (≥75 yo) = 13255, of which metastatic CRC: 3588 (19.5%) Younger (<75 yo) = 18410, of which metastatic CRC: 2724 (20.5%) | **Metastatic site:** most frequently the liver (68% in the Younger group vs. 72% in the Older group), followed by the peritoneum (33%), lung (21%), bones (5%) and brain (1%) **Synchronous metastasis: 2724** patients in the Older vs. 3588 in Younger | Older: < 50% received palliative CT, Younger: 85% | Not reported | Not reported | Not reported | OS of patients with metastatic CRC: Older: 8.4 months (95%CI: 7.6–9.4) Younger: 22.3 months (95%CI: 21–24.9) | Not reported | • **Age <85 years, isolated metastasis, no bowel obstruction and Charlson Comorbidity Index ≤2, CT, liver surgery, and primary tumor resection were significant predictors of improved OS** |
| **Booth et al. 2015** | Population-based retrospective cohort study 2002–2009 | **Liver resection** Including major (34%) and minor (64%) hepatectomies | 1310 Older (≥75 yo) = 186 Younger (65–74 yo) = 414 Young (<65) = 710 | Not reported | Older: 14% Younger: 15% Young: 16% Peri-operative CT was less common in Older patients: 41% of Older vs. 57% of Young er, vs. 71% of young patients (p<0.001). | Not reported | At 30 days: Older: 5% Younger: 3% Young: 1% (p = 0.005) At 90 days: Older: 8% Younger: 5% Young: 2% (p<0.001) | Not reported | 5-year OS: Older: 28% Younger: 44% Young: 49% (p<0.001). 10-year OS: Older: 12% Younger: 23% Young: 35% (p<0.001) | Not reported | • **Increasing age and major hepatectomy were independent predictors of mortality at 30 days** • Resection of CRLM is associated with greater risk of postoperative mortality among Older patients despite less aggressive treatment. |
| **Nomi et al. 2015** | Retrospective single center matched case control study 1998–2013 | **Liver resection** All resections were performed with curative intent | 93 Older (≥70 yo) = 31 Younger (<70 yo) = 62 | **CRLM size ≥ 5 cm:** 32.3% of Older vs. 29% of Younger **Median number of lesions:** 2 (range: 1–8) for Olders vs. 2 (1–6) for Younger (NSD) | Older: 58.1% Younger: 75.8% (p = 0.09) | Older: 41.9% Younger: 54.8% (p = 0.276) | At 90 days: 0% for both groups | Older: 83.9% Younger: . 95.2% (p = 0.116) | 3-year OS: Older: 57.9% Younger: 61.7% (p = 0.842) | 3-year DFS: Older: 38.5% Younger: 35.3% (p = 0.676) | • Laparoscopic major hepatectomy for CRLM could be safely performed in Older patients • **Advanced age itself should not be regarded as contraindication for liver surgery** |

(*Continued*)

Table 1. (Continued)

| Ref. | Study design and time frame | Type of intervention | Number of patients | Metastasis characteristics | Neoadjuvant chemotherapy | Morbidity rate | Mortality rate | R0 resection | OS | DFS | Predictors of morbidity and survival |
|---|---|---|---|---|---|---|---|---|---|---|---|
| **Parakh et al. 2015** | Population-based retrospective cohort study 2009–2014 | Multi treatments: **Surgical resection of metastatic disease occurred in 21% of patients**, declining with advancing age (26% in Younger vs. 21% in Older vs. 6% in very old, p<0.001) | 821 Very old (≥85 yo) = 106 Older (75–84 yo) = 352 Younger (65–74) = 363 | **Metastatic site:** liver (62%), lung (32%), brain (3%), bone (1.5%) **Synchronous metastatic disease:** 58% patients No age-related differences | Overall 23% of patients, with differences between age groups (only 34% of very old received neoadjuvant CT) | Not reported | Not reported | Not reported | Median OS: Very old:11 months Older: 20 months Younger: 26 months (p<0.001) | Not reported | • Longer median survival was observed for patients who received CT across each of the age groups, though not reaching statistical significance in those ≥85 years (p = 0.061) • >**Older patients (aged 75–84) (HR: 1.33, 95%CI: 1.09–1.63) and very old (aged ≥ 85 years) (HR: 2.39, 95%CI: 1.80–3.16) had a poorer OS** than Younger patients (aged 65–74 years) |
| **Nachmany et al. 2016** | Retrospective cohort study 2010–2015 | **Liver resection** 25% by laparoscopy (22.5% in the Older vs. 27.7% in the Younger, p = 0.49). Major liver resection: 20.3% in Older vs. 34.1% in Younger (NSD) | 174 Older (≥70 yo) = 54 Younger (<70 y.o.) = 120 | **Number of metastasis:** 1.7 (SD: 1.38) for Older vs. 2.96 (SD: 2.86) for Younger (p = 0.003) **Maximal lesion size:** 37 mm (SD: 25.9) for Older vs. 32.9 mm (SD: 26.3) for Younger (p = 0.18) **Bilobar disease:** 16.6% for Older vs. 33.3% for Younger (p = 0.07) | Older: 55.5% Younger: 76.6% (p = 0.13) | Older : 11.1% Younger: 2.5% r (p<0.0001) | At 60 days: Older: 1.8% | Older: 90% Younger: 86% (p = 0.14) | 3-year OS: NSD | 3-year DFS: NSD | Not reported |
| **Nardo et al. 2016** | Retrospective multicenter cohort study 2008–2015 | Liver resection | 149 Very old (≥75 yo) = 21 Older (65–74 yo) = 79 Younger (<65) = 49 | **Unilobar liver metastases:** 16.7% in the very old group vs. 21.1% in the Older vs. 28.6% in the Younger **Mean lesion size:** 4.6 cm for very old vs. 4.5 cm for Older vs. 4.1 cm for Younger (NSD) | Very old: 14.3% Older: 16.5% Younger: 16.3% (p = 0.97) | Very old: 24.1% Older: 24% Younger: 22.4% (p = 0.86) | At 30 days: Very old: 4.8% Older: 2.5% Younger: 0% (p = 0.8) At 60 days: Very Old: 4.8% Older: 3.8% Younger: .2% (p = 0.8) | Very old: 90.5% Older: 90.1% Younger: 93.9% (p = 0.83) | 1-, 3- and 5-year OS: Very old: 85.7%, 38.9%, 28.6% Older: 89.9%, 38%, 33.3% Younger: 87.6%, 53.5%, 43.5% (NSD) | 1-, 3- and 5-year DFS: Very old: 76.2%, 31.3%, 20% Older: 75.9%, 35%, 28.6% Younger: 77.1%, 37.6%, 36.4% (NSD) | • **Advanced chronological age cannot be considered a medical or surgical contraindication to hepatic resection for CRLM** |
| **Gandy et al. 2018** | Retrospective cohort study 2007–2014 | **Liver resection** The majority of patients in both groups (64% and 62%) underwent major liver resections (>3 segments) | 187 Older (≥75 yo) = 29 Younger (<75 y.o.) = 158 | Not reported | Older: 52% Younger: 69% (p = 0.71) | Older:13.8% Younger: 16.5% (p = 0.65) | 1 patients per group (NSD) | Not reported | 1-, 3-, 5-year OS: Older: 92.3%, 67.3%, 57.7% Younger: 95.1%, 72.9%, 55.6% (p = 0.6) | Not reported | Not reported |

*(Continued)*

**Table 1.** (Continued)

| Ref. | Study design and time frame | Type of intervention | Number of patients | Metastasis characteristics | Neoadjuvant chemotherapy | Morbidity rate | Mortality rate | R0 resection | OS | DFS | Predictors of morbidity and survival |
|---|---|---|---|---|---|---|---|---|---|---|---|
| **Yue et al. 2018** | Retrospective single center cohort study 2009–2016 | **Liver resection** All laparoscopic hepatectomiesincluding left lateral sectionectomy, sectionectomy, wedge resections | 241 Older (≥70 yo) = 78 Younger 60–69 y.o.) = 163 | **Largest lesion size:** 2 cm (1–4) for Older vs. 3 cm (1–5) for Younger (p = 0.128) **Number of lesions:** 2 (1–3) for Older vs. 2 (1–4) for Younger (p = 0.20) | Older: 88.4% Younger: 72.3% (p = 0.005) | At 90 days: Older: 26.9% Younger: 23.3% (p = 0.5) | At 90 days: 1 patient per group (overall: 0.8%) | 100% for both groups | 5-year OS: Older: 52% Younger: 59%, (p = 0.139) | 5-year DFS: Older: 45% Younger: 49% (p = 0.09) | • TNM stage, disease-free interval, and number of metastases were independent predictors of OS • Disease-free interval and preoperative carcinoembryonic antigen levels were independent predictors of DFS • Age did not independently predict OS or DFS |
| **Zarzavadjian Le Bian et al. 2019** | Retrospective cohort study 2008–2017 | **Liver resection** All laparoscopic procedures Local radiotherapy was performed in 34% of rectal cancers | 335 Very old (>75 yo) = 52 Older (65–75 yo) = 136 Younger (55–65 yo) = 113 Young (< 55 yo) = 34 | **Synchronous liver metastases:** 154 patients **Bilobar metastasis:** 107 patients **More than 5 CRLM:** 19 patient | Overall, 47.5% of patients | Very old: 9.6% Older: . 7.4% Younger: 12.4% Young: 17.6% (p = 0.287) | At 90 days: 0% | Not reported | Not reported | Not reported | Not reported |

NSD stands for non significantly different.

**Table 2. Summary of study characteristics and study outcomes of the articles comparing different age groups of patients (older vs. younger) receiving non-surgical local ablation treatments (including radioembolisation, radiofrequency ablation, and high-dose-rate brachytherapy) for metastatic colorectal cancer.**

| Ref. | Study design and time frame | Study population | Type of intervention | N | Response to treatment (RECIST) | Morbidity | Mortality | Overall Survival | Predictors of morbidity and survival |
|---|---|---|---|---|---|---|---|---|---|
| **Tohme et al. 2014** | Retrospective cohort study 2002–2012 | Consecutive mCRC patients non candidates for surgery and treated with radioembolisation after multiples CT regimens or refusing standard CT | **Radioembolization** with yttrium-90 (90Y)-labeled resin microspheres (90Y radioembolization [90Y-RE]) Lobar approach | 107 Older (≥70 yo) = 44 Younger (<70 y.o.) = 64 | 1.1 up to 6 months after RE | RE was equally well tolerated in both groups. | At 90 days: Older: 13.6% Younger: 12.5% | Median survival: Older: 8.2 months (95%CI: 5.9–10.5) Younger: 8.4 months (95%CI: 6.2–10.6) (p = 0.351) | • For the entire cohort, **extrahepatic disease at the time of treatment was the only independent predictor of worse OS,** but not confirmed in the multivariate analysis. • **Age alone should not be a discriminating factor for the use of 90Y-RE** in the management of mCRC patients. |
| **Kennedy et al. 2015** | Retrospective multicenter cohort study 2002–2011 | Consecutive patients receiving radioembolization for advanced liver-dominant mCRC who were not suitable for surgery, ablation, or systemic therapy or declined consent | **Radioembolization** with yttrium-90 (90Y)-labeled resin microspheres (90Y radioembolization [90Y-RE]) Lobar approach for 43% of patients | 606 Older (≥70 yo) = 160 Younger (<70 y.o.) = 446 | 1.0 and 1.1 at 3 months after radioembolisation | CTCAE Grade 1–2: Older: 48.1% Younger: 48.6% CTCAE Grade > = 3: Older: 16.9% for Younger: 18.8% (p = 0.433) | At 30 days: Older: 1.9% Younger: 2.0% (p = 1) At day 60: Older: 6.3% Younger: 6.1% (p = 1) At 90 days: Older: 18.1% Younger: 12.6% (p = 0.086) | Median OS: Older: 9.3 months (95%CI: 8–12.1) Younger: 9.7 months (95%CI: 9–11.4) (p = 0.335) | • **Age was not a factor in determining the treatment approach for 90Y-RE,** but Older patients were less likely to receive more than 1 90Y-RE procedure (p = 0.007), and a lower volume of liver was treated (p<0.001) |
| **Seidensticker et al. 2018** | Retrospective single center cohort study 2006–2010 | mCRC patients receiving at least one RFA, Y90-RE, or HDR-BT after failure of CT and surgical treatment | **Radiofrequency ablation (RFA), high-dose-rate brachytherapy (HDR-BT), or Y90-radioembolization (Y90-RE)** | 266 RFA = 60 HDR-BT = 192 Y90-RE = 96 Older (≥70 yo) = 89 Younger (<70 y.o.) = 177 | Not reported | Not reported | Not reported | Median OS: Older: 16.6 months Younger: 13.2 months (p = 0.19) *By treatment option:* For RFA: Older: 26.7 months Younger: 24.3 (p = 0.76) For HDR-BT: Older: 19.1 months Younger: 18.2 months (p = 0.83) For Y90-RE: Older: 6.9 months Younger: 6.5 months (p = 0.86) | **The type of local ablation treatment had no impact on OS in older patients** |

CTCAE: Common Terminology Criteria for Adverse Events (version 3)

**Table 3. Summary of study characteristics and study outcomes of the articles comparing different types of intervention for metastatic colorectal cancer in older patients.**

| Ref. | Study design and time frame | Types of intervention | N | Age Mean (SD) or Median (range) | Metastasis characteristics | Morbidity | Mortality | R0 resection | Overall Survival (OS) | Disease-Free Survival (DFS) | Predictors of morbidity and survival |
|---|---|---|---|---|---|---|---|---|---|---|---|
| **Zacharias et al. 2004** | Retrospective cohort study 1990–2000 | **First hepatectomy (FH) vs. Repeated hepatectomy (RH)** | 61 56 FH vs. 14 RH | 73 (70–81) | **Synchronous metastasis: 36%** in FH vs. 0% in RH **Mean lesion number:** 2 in FH vs. 2 in RH **Unilobar metastasis:** 71% in FH vs. 79% in RH **Mean lesion diameter:** 6 cm in FH vs. 7 cm in RH | FH group: 36 complications in 22 patients. For RH group: 7 complications in 4 patients | FH: 0% RH: 7% | FH: 53 patients | 1,3,5 years OS: FH: 86%, 44%, 21% RH: 61%, 25% No survivor at 5 years | 1,2,3, 5 years DFS: FH: 45%, 23%, 19% Median DFS: 12 months. | • CEA level > 200 ng/mL and ≥3 liver metastases were independent risk factors for poor OS. • Presence of extrahepatic disease, ≥3 liver metastases, and CEA level > 200 ng/mL were independent risk factors for recurrence and poor DFS |
| **Cummings et al. 2007** | Population-based retrospective cohort study 1991–2001 | **Hepatic resection (HR) vs. no hepatic resection (No HR)** | 13599 833 HR vs. 12766 No HR | For HR: 71.6 (5.1) years for LM and 73.8 years (6.4) for DM group For No HR: 75.6 (6.8) years for LM and 75.8 (6.8) years for DM group | **Local metastasis (LM):** 5926 patients **Distant metastasis (DM):** 7673 patients | HR: 384 complications in 263 patients, | HR: 4.3% | Not reported | 5-year OS: HR: 32.8% No HR: 10.5% (p<0.0001) | Not reported | • **HR was associated with improved survival** in the LM and DM groups: not undergoing HR was associated with a 1.9-fold increase in the risk of death • **Age was associated with an increased risk of death.** Each increase in age by 1 year increased the risk of death by 3%. |
| **Khan et al. 2014** | Population-based retrospective cohort study 2004–2012 | **Hepatic resection (HR) vs. no hepatic resection (No HR)** | 41137 36109 HR (20617 aged ≥65 y.o.) vs. 5028 No HR (2257 aged ≥65 y.o.) | For HR: 67 (57–78) years For No HR: 63 (53–73) years | Not reported | Not reported | Not reported | Not reported | 3-year OS: HR:: 33.9%, No HR: 16.8% | Not reported | • **OS was highly dependent on age** • **OS was significantly better in patients who underwent HR** • **Favorable association between HR and OS up to 85 years old** |
| **Grande et al. 2016** | Retrospective multicenter cohort study 2000–2013 | **CT vs. others treatments (including palliative care but No CT)** | 751 Patients receiving CT (alone or combined with surgery or radiotherapy) = 57 Patients receiving surgery (3%) or palliative cares (21%) = 173 | 79 (75–93) | **Synchronous metastasis:** 58.5% **Liver metastases only:** 41.1% **Lung metastasis only:** 10.3% **Multi-organ metastases:** 34.4% **Other site:** 1.2% | Not reported | Not reported | Not reported | Median OS: 17 months (CI95%: 15–19) For No CT group: 5 months (4–6) For CT group: 20 months (18–22) For patients with only metastasis resection (n = 19): 22 months For patients ≥80 y.o.: OS was 17.6% for No CT group, and 34.8% for CT group (P>0.0001) | Not reported | • Sex (female), age (<80y), performance status (ECOG-PS: 0–1), chemotherapy, surgery of metastasis, surgery of primary tumor and site of metastasis (liver) were prognostic factors for OS. |

**Table 3.** (Continued)

| Ref. | Study design and time frame | Types of intervention | N | Age Mean (SD) or Median (range) | Metastasis characteristics | Morbidity | Mortality | R0 resection | Overall Survival (OS) | Disease-Free Survival (DFS) | Predictors of morbidity and survival |
|---|---|---|---|---|---|---|---|---|---|---|---|
| **Massarweh et al. 2016** | Population-based retrospective cohort study 1998–2009 | **Primary tumor resection (PTR) vs. MMT vs. CT** | 103100 PTR alone = 29841 MMT = 44247 CT alone = 13979 No treatment = 15033 | PTR: 71.1 ± 13.0 years MMT: 61.2 ± 12.8 years CT: 62.5 ± 13.1 years No treatment: 72.4 ± 12.8 | Not reported | Not reported | Not reported | Not reported | Not reported | Not reported | • **Survival estimates decreased with increasing patient age,** but the pattern of survival across treatment strategies was similar. <br>• After adjusting for relevant covariates, **risk of death was significantly higher for all treatment strategies compared with MMT** (no treatment: hazard ratio [HR] 1.74 [1.60–1.88]; CT: HR 1.86 [1.77–1.95]; PTR alone: HR 1.21 [1.16–1.26]). **Among those aged 65–74 years (HR 0.76 [0.70–0.83]) and ≥75 years (HR 0.76 [0.71–0.82]), MMT was associated with a significantly lower risk of death compared to PTR alone** |
| **Zeng et al. 2016** | Retrospective case control study with PSM 2008–2016 | **Laparoscopic hepatectomy (LH) vs. Open hepatectom (OH)** | 385 After PSM: LH = 79 OH = 79 | LH: 69 (65–75) years OH: 68 (65–76) years | **Mean size:** 2.9 cm for LH vs. 3.3 cm for OH **Right lesions:** 31 LH vs. 36 OH **Left lesions:** 48 Lh vs. 43 OH | LH : 17.7% OH : 24% (p = 0.328) | 0% | Not reported | 5-year OS: 51% (p = 0.276) | 5-year DFS: 42% (p = 0.49) | • **The disease-free interval was a significant predictor of OS.** <br>• **The operative approach was not a predictor** of 5-year OS and 5-year DFS. |
| **Martinez Cecilia et al. 2017** | Retrospective multicenter case control study with PSM 2005–2012 | **Laparoscopic hepatectomy (LH) vs. Open hepatectom (OH)** | 775 After PSM: LH = 225 OH = 225 | LH; 75 (70–87) years OH: 75 (70–86) years | **Unilobar liver metastasis:** 80% LH vs. 82% OH **Median largest lesion:** 30 mm for LH vs. 30 mm for OH | LH : 22% OH : 39% (p<0.001) | Not reported | Not reported | 1-, 3-, 5-year OS: LH: 93%, 68%, 43% OH: 89% 60%, 46% | 1-, 3-, 5-year DFS: LH: 71%, 43%, 31% OH: 75%, 46%, 29% | • **70–74 y.o. subgroup:** overall morbidity was significantly lower in LH compared with OH (17% vs 41%, P<0,001). OS was 52 (95% CI 36–67) months in the LH and 43 (95% CI 21–65) months in the OH (p = 0.425) <br>• **75–79 y.o. subgroup:** overall morbidity was significantly lower in LH compared with OH (23% vs 47%, p<0,006). NSD for survival rates. <br>• **Over 80 y.o. subgroup:** NSD for survival and postop morbidity between LH and OH |

COPD = chronic obstructive pulmonary disease; CRLM = colorectal liver metastasis; CT = chemotherapy; MMT = multimodality treatments; NR = not reported; NSD = not significant difference; PSM = propensity score matching; SEER = Surveillance, Epidemiology and End Results database

rates [25, 40, 42, 44]. Similarly, 12 studies reported a similar postoperative mortality between older and younger patients [45, 47, 50–52, 55–57, 61–64], whereas a significantly higher post-operative mortality was observed in 4 studies [40, 54, 59, 60]. In the majority of the studies (11/18, 61.1%), older patients had a worse overall survival compared to younger patients [25, 40, 42, 44, 47, 54, 55, 57, 59, 62]. However, the majority of the studies found that age was not an independent predictor of OS and DFS, supporting the conclusion that the advanced chronologic age should not be regarded as a medical contraindication to hepatic resection for CRLM.

Meta-analyses were performed by the patients' age group and as a global comparison between older and younger patients. Eleven studies compared patients aged ≥70 years vs. patients aged <70 years [47, 50, 52, 54–57, 59, 62–64], and 5 studies were selected to compare patients aged ≥75 years vs. patients aged <75 years [40, 45, 51, 60, 61].

The operative time was significantly shorter in older patients, whereas no age-related difference was found for the transfusion rate and R0 resection (Fig 2). Blood loss was estimated in only 6 studies comparing 337 patients aged ≥70 years vs. 839 patients aged <70 years [50, 52, 56, 57, 62, 64]. Pooled data analysis showed a non significant difference between the groups (MD: 14.85 [95%CI: -8.59; 38.27], p = 0.21; $I^2$: 44%) (S2 Fig).

Postoperative bile leak and liver failure were not different between older and younger patients undergoing liver resection for CRLM (S3 Fig). Pulmonary complications were evaluated in 6 studies comparing 397 patients aged ≥70 years vs. 1096 patients aged <70 years [54, 56, 57, 62–64] and were not different between the groups (RR: 1.75 [95%CI–0.69; 4.44], p = 0.24; $I^2$: 66%) (S4 Fig). The rates of overall postoperative complications (Fig 3A), as well as the occurrence of major postoperative complications (classified as Dindo-Clavien III or more) (S5 Fig) and the overall hospital stay (Fig 3B), were similar between older and younger patients. Conversely, a significantly higher postoperative mortality was observed in older patients (RR: 2.53), overall and when considering age cut-off of ≥70 years old or ≥75 years old (Fig 3C). Concerning the survival analysis, older patients had a worse survival compared with younger patients but a similar DFS (Fig 4).

## Outcomes of non-surgical local ablation of mCRC in older vs. younger patients

The outcomes of yttrium-90-labeled resin microsphere radioembolization (90Y-RE) were assessed in 3 studies involving a total of 809 patients [49, 53, 58] and were compared between older (≥70 years old) and younger patients (<70 years old) (Table 2). 90Y-RE was indicated for patients with CRLM who were refractory, exhausted or declined standard CT regimens [49, 58], or patients with diffuse, liver-dominant involvement [53]. Overall, 90Y-RE was equally well tolerated in both the older and younger patient groups, with no different rate and severity of adverse events. Mortality at 90 days ranged between 12.5% to 18.1%, without differences related to the age group [49, 58]. Median survival did not reach 1 year after 90Y-RE (range: 6.5–9.7 months)[49, 53, 58] independent of the age group. Although older patients appeared to be less likely to receive more than 1 90Y-RE procedure and a lower liver volume was treated, all 3 studies concluded that age alone should not be a discriminating factor for the use of 90Y-RE in the management of mCRC [49, 53, 58]. Pooled data analysis was possible only for postoperative mortality. Based on two studies [49, 58], including 204 older patients vs. 510 younger patients, the postoperative mortality rate post 90Y-RE was not different between the groups (RR: 1.39 [95%CI: 0.95, 2.02], p = 0.09; $I^2$: 0%) (S6 Fig).

Only one study evaluated the outcomes of radiofrequency ablation (RFA) applied in 60 mCRC patients, and high-dose-rate brachytherapy (HDR-BT) applied in 192 mCRC patients

## a. Operative Time

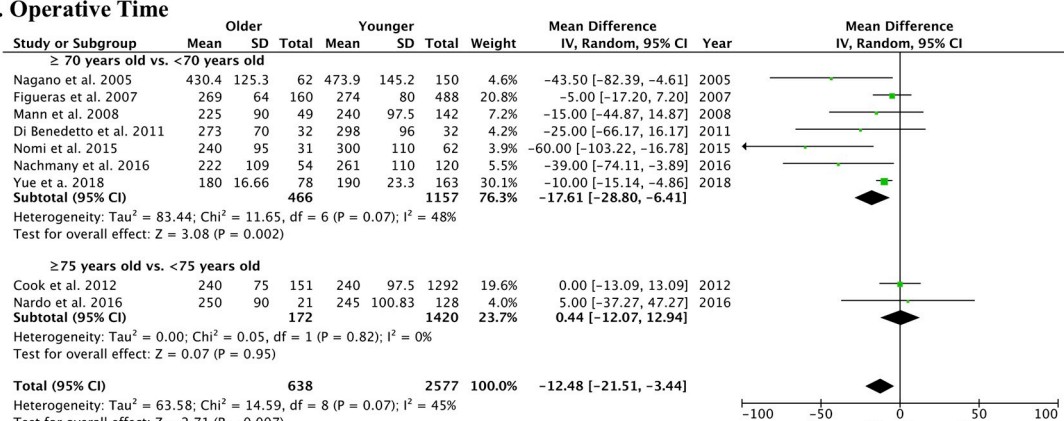

## b. Transfusion

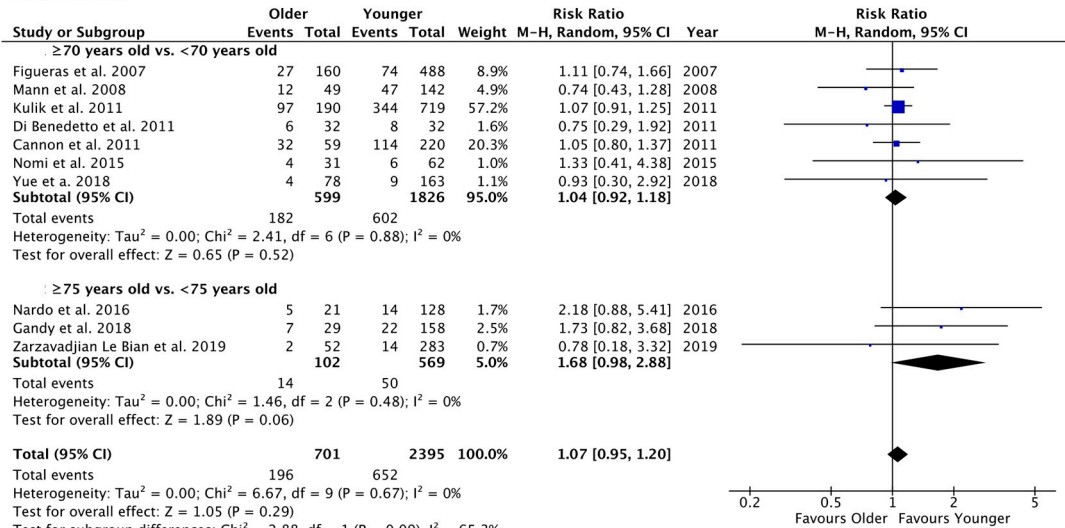

## c. R0 Resection

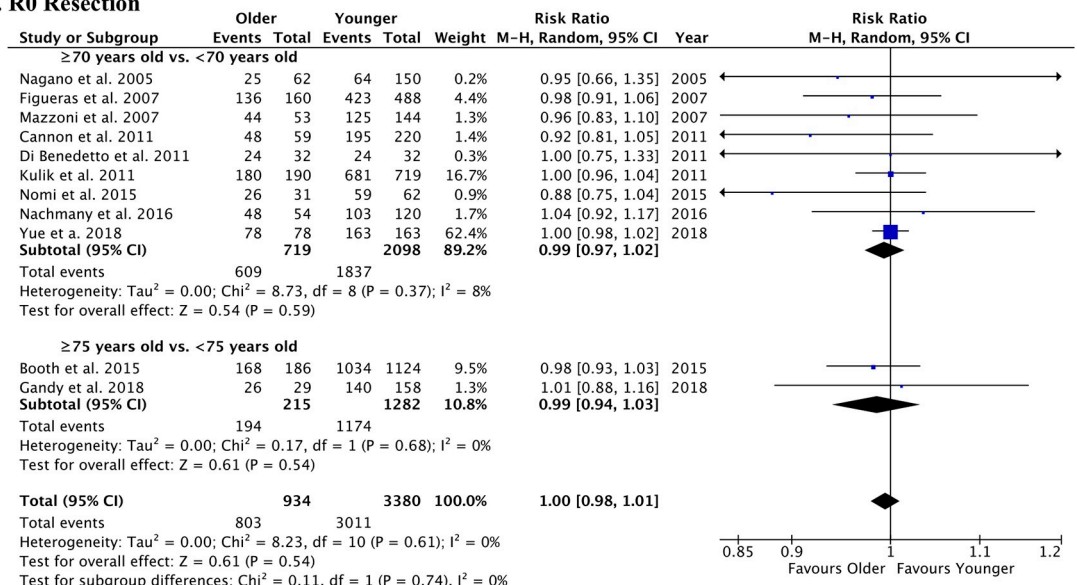

**Fig 2. Forest plots of operative outcomes of liver resection for CRLM in older vs. younger patients.** The following outcomes were analyzed: a. operative time (min); b. transfusion rate (n); and c. R0 resection (n).

[53]. For RE, local ablation was selected in potentially resectable metastases only if patients had an unfavorable performance status and/or severe comorbidities (resulting in a high risk of perioperative morbidity and mortality) or if patients refused surgery. RFA was preferentially applied for patients with single lesions up to 3 cm in diameter, whereas interstitial HDR-BT was applied for oligometastatic disease [53]. The median survival rate reported for RFA was 26.7 months for older patients vs. 24.3 months for younger patients, whereas it was 19.1 months for older patients vs. 18.2 months for younger patients receiving HDR-BT. No age-related differences were found, suggesting that local ablation treatments can be safely performed in older people, although the type of local ablation treatment seems to have no impact on OS. The presence of comorbidity, in particular moderate to severe renal insufficiency, appeared to negatively impact the outcomes of local therapies [53].

## Outcomes of different treatments for mCRC in older patients

Comparisons at the intervention level included 1 study that compared first hepatectomy vs. repeated hepatectomy [46] in older patients ($\geq$70 years old), 2 studies that compared hepatic resection vs. no hepatic resection for CRLM [26, 41], 2 studies that compared CT vs. other treatments (including surgery and multimodality treatments, MMT)[43, 48], and 2 studies that compared laparoscopic hepatectomy vs. open hepatectomy [65, 66] in older patients (Table 3). CRC metastases were highly heterogeneous in their presentation, in terms of number of lesions, size and type (synchronous or metachronous). Overall, survival was dependent on the patient's age [26, 41, 43, 48] and was significantly better in patients who underwent hepatic resection for CRLM (based on 2 population-based studies, including 54736 patients)[26, 41].

The comparisons between laparoscopic and open surgery for the resection of CRLM in older patients (2 PSM studies, including 1160 patients before PSM and 608 patients after PSM) showed that the operative approach is not a predictor of 5-year OS and 5-year DFS [65, 66], although a significantly lower postoperative morbidity was associated with laparoscopic hepatectomy, particularly in the age groups < 80 years [66]. The meta-analytic approach was used to pool together data from these two studies and compare laparoscopy vs. open surgery [65, 66], reaching a total of 304 patients in the laparoscopic group and 304 patients in the open surgery group. The forest plots (Fig 5) show a significant difference in favor of laparoscopy for blood loss, hospital stay, and rate of major postoperative complications. The operative time was not different between the two surgical approaches. A sensitivity analysis was conducted pooling together data from the study by Zeng et al. [65](all patients aged $\leq$75 years) and a sub-sample of patients from the study of Martinez-Cecilia et al. [66] aged between 70 and 74 years (n = 356). This analysis confirmed the previous results for operative time (MD: 13.9 [95%CI: -39.93, 67.73]; p = 0.61; $I^2$: 95%), blood loss (MD: -72.54 [95%CI: -100.19, -44.89]; p<0.001; $I^2$: 93%), hospital stay (MD: -2.86 [95%CI: -4.55, 0.55]; p<0.0001; $I^2$: 0%), and severe postoperative complication rate (RR = 0.27 [95%CI: 0.10, 0.73]; p = 0.01; $I^2$: 0%) between laparoscopy and open surgery. OS and DFS were not different between the two surgical approaches (Fig 5E and 5F).

## Study quality assessment

Fourteen studies were judged as having low quality and thus had a high risk of bias (S1 Table). The remaining studies scored $\geq$ 6 on the NOS.

## a. Postoperative Complications

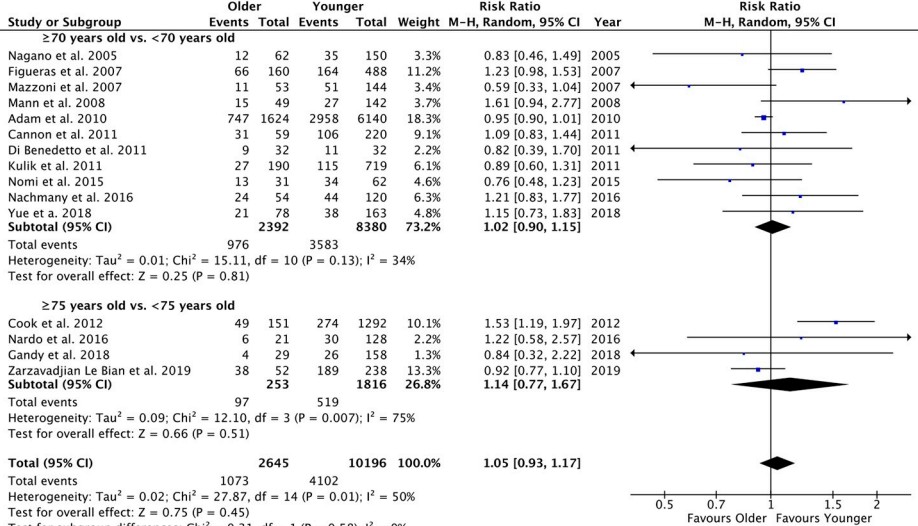

| Study or Subgroup | Older Events | Total | Younger Events | Total | Weight | Risk Ratio M-H, Random, 95% CI | Year |
|---|---|---|---|---|---|---|---|
| **≥70 years old vs. <70 years old** | | | | | | | |
| Nagano et al. 2005 | 12 | 62 | 35 | 150 | 3.3% | 0.83 [0.46, 1.49] | 2005 |
| Figueras et al. 2007 | 66 | 160 | 164 | 488 | 11.2% | 1.23 [0.98, 1.53] | 2007 |
| Mazzoni et al. 2007 | 11 | 53 | 51 | 144 | 3.4% | 0.59 [0.33, 1.04] | 2007 |
| Mann et al. 2008 | 15 | 49 | 27 | 142 | 3.7% | 1.61 [0.94, 2.77] | 2008 |
| Adam et al. 2010 | 747 | 1624 | 2958 | 6140 | 18.3% | 0.95 [0.90, 1.01] | 2010 |
| Cannon et al. 2011 | 31 | 59 | 106 | 220 | 9.1% | 1.09 [0.83, 1.44] | 2011 |
| Di Benedetto et al. 2011 | 9 | 32 | 11 | 32 | 2.2% | 0.82 [0.39, 1.70] | 2011 |
| Kulik et al. 2011 | 27 | 190 | 115 | 719 | 6.1% | 0.89 [0.60, 1.31] | 2011 |
| Nomi et al. 2015 | 13 | 31 | 34 | 62 | 4.6% | 0.76 [0.48, 1.23] | 2015 |
| Nachmany et al. 2016 | 24 | 54 | 44 | 120 | 6.3% | 1.21 [0.83, 1.77] | 2016 |
| Yue et a. 2018 | 21 | 78 | 38 | 163 | 4.8% | 1.15 [0.73, 1.83] | 2018 |
| **Subtotal (95% CI)** | | 2392 | | 8380 | 73.2% | 1.02 [0.90, 1.15] | |
| Total events | 976 | | 3583 | | | | |

Heterogeneity: Tau² = 0.01; Chi² = 15.11, df = 10 (P = 0.13); I² = 34%
Test for overall effect: Z = 0.25 (P = 0.81)

| | | | | | | | |
|---|---|---|---|---|---|---|---|
| **≥75 years old vs. <75 years old** | | | | | | | |
| Cook et al. 2012 | 49 | 151 | 274 | 1292 | 10.1% | 1.53 [1.19, 1.97] | 2012 |
| Nardo et al. 2016 | 6 | 21 | 30 | 128 | 2.2% | 1.22 [0.58, 2.57] | 2016 |
| Gandy et al. 2018 | 4 | 29 | 26 | 158 | 1.3% | 0.84 [0.32, 2.22] | 2018 |
| Zarzavadjian Le Bian et al. 2019 | 38 | 52 | 189 | 238 | 13.3% | 0.92 [0.77, 1.10] | 2019 |
| **Subtotal (95% CI)** | | 253 | | 1816 | 26.8% | 1.14 [0.77, 1.67] | |
| Total events | 97 | | 519 | | | | |

Heterogeneity: Tau² = 0.09; Chi² = 12.10, df = 3 (P = 0.007); I² = 75%
Test for overall effect: Z = 0.66 (P = 0.51)

| | | | | | | | |
|---|---|---|---|---|---|---|---|
| **Total (95% CI)** | | 2645 | | 10196 | 100.0% | 1.05 [0.93, 1.17] | |
| Total events | 1073 | | 4102 | | | | |

Heterogeneity: Tau² = 0.02; Chi² = 27.87, df = 14 (P = 0.01); I² = 50%
Test for overall effect: Z = 0.75 (P = 0.45)
Test for subgroup differences: Chi² = 0.31, df = 1 (P = 0.58), I² = 0%

## b. Hospital Stay

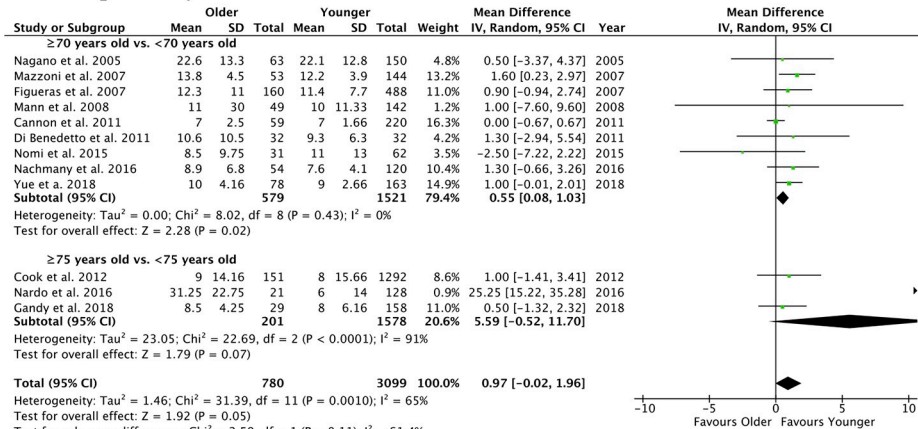

| Study or Subgroup | Older Mean | SD | Total | Younger Mean | SD | Total | Weight | Mean Difference IV, Random, 95% CI | Year |
|---|---|---|---|---|---|---|---|---|---|
| **≥70 years old vs. <70 years old** | | | | | | | | | |
| Nagano et al. 2005 | 22.6 | 13.3 | 63 | 22.1 | 12.8 | 150 | 4.8% | 0.50 [-3.37, 4.37] | 2005 |
| Mazzoni et al. 2007 | 13.8 | 4.5 | 53 | 12.2 | 3.9 | 144 | 13.2% | 1.60 [0.23, 2.97] | 2007 |
| Figueras et al. 2007 | 12.3 | 11 | 160 | 11.4 | 7.7 | 488 | 11.0% | 0.90 [-0.94, 2.74] | 2007 |
| Mann et al. 2008 | 11 | 30 | 49 | 10 | 11.33 | 142 | 1.2% | 1.00 [-7.60, 9.60] | 2008 |
| Cannon et al. 2011 | 7 | 2.5 | 59 | 7 | 1.66 | 220 | 16.3% | 0.00 [-0.67, 0.67] | 2011 |
| Di Benedetto et al. 2011 | 10.6 | 10.5 | 32 | 9.3 | 6.3 | 32 | 4.2% | 1.30 [-2.94, 5.54] | 2011 |
| Nomi et al. 2015 | 8.5 | 9.75 | 31 | 11 | 13 | 62 | 3.5% | -2.50 [-7.22, 2.22] | 2015 |
| Nachmany et al. 2016 | 8.9 | 6.8 | 54 | 7.6 | 4.1 | 120 | 10.4% | 1.30 [-0.66, 3.26] | 2016 |
| Yue et a. 2018 | 10 | 4.16 | 78 | 9 | 2.66 | 163 | 14.9% | 1.00 [-0.01, 2.01] | 2018 |
| **Subtotal (95% CI)** | | | 579 | | | 1521 | 79.4% | 0.55 [0.08, 1.03] | |

Heterogeneity: Tau² = 0.00; Chi² = 8.02, df = 8 (P = 0.43); I² = 0%
Test for overall effect: Z = 2.28 (P = 0.02)

| | | | | | | | | | |
|---|---|---|---|---|---|---|---|---|---|
| **≥75 years old vs. <75 years old** | | | | | | | | | |
| Cook et al. 2012 | 9 | 14.16 | 151 | 8 | 15.66 | 1292 | 8.6% | 1.00 [-1.41, 3.41] | 2012 |
| Nardo et al. 2016 | 31.25 | 22.75 | 21 | 6 | 14 | 128 | 0.9% | 25.25 [15.22, 35.28] | 2016 |
| Gandy et al. 2018 | 8.5 | 4.25 | 29 | 8 | 6.16 | 158 | 11.0% | 0.50 [-1.32, 2.32] | 2018 |
| **Subtotal (95% CI)** | | | 201 | | | 1578 | 20.6% | 5.59 [-0.52, 11.70] | |

Heterogeneity: Tau² = 23.05; Chi² = 22.69, df = 2 (P < 0.0001); I² = 91%
Test for overall effect: Z = 1.79 (P = 0.07)

| | | | | | | | | | |
|---|---|---|---|---|---|---|---|---|---|
| **Total (95% CI)** | | | 780 | | | 3099 | 100.0% | 0.97 [-0.02, 1.96] | |

Heterogeneity: Tau² = 1.46; Chi² = 31.39, df = 11 (P = 0.0010); I² = 65%
Test for overall effect: Z = 1.92 (P = 0.05)
Test for subgroup differences: Chi² = 2.59, df = 1 (P = 0.11), I² = 61.4%

## c. Postoperative Mortality

| Study or Subgroup | Older Events | Total | Younger Events | Total | Weight | Risk Ratio M-H, Random, 95% CI | Year |
|---|---|---|---|---|---|---|---|
| **≥70 years old vs. <70 years old** | | | | | | | |
| Nagano et al. 2005 | 0 | 62 | 1 | 150 | 0.6% | 0.80 [0.03, 19.35] | 2005 |
| Mazzoni et al. 2007 | 3 | 53 | 3 | 144 | 2.3% | 2.72 [0.57, 13.05] | 2007 |
| Figueras et al. 2007 | 12 | 160 | 13 | 488 | 9.5% | 2.82 [1.31, 6.04] | 2007 |
| Mann et al. 2008 | 2 | 49 | 4 | 142 | 2.0% | 1.45 [0.27, 7.67] | 2008 |
| Adam et al. 2010 | 62 | 1624 | 101 | 6140 | 54.8% | 2.32 [1.70, 3.17] | 2010 |
| Di Benedetto et al. 2011 | 1 | 32 | 0 | 32 | 0.6% | 3.00 [0.13, 71.00] | 2011 |
| Kulik et al. 2011 | 1 | 190 | 9 | 719 | 1.3% | 0.42 [0.05, 3.30] | 2011 |
| Cannon et al. 2011 | 0 | 59 | 9 | 220 | 0.7% | 0.19 [0.01, 3.28] | 2011 |
| Nomi et al. 2015 | 0 | 31 | 0 | 62 | | Not estimable | 2015 |
| Nachmany et al. 2016 | 1 | 54 | 0 | 120 | 0.6% | 6.60 [0.27, 159.46] | 2016 |
| Yue et a. 2018 | 1 | 78 | 1 | 163 | 0.7% | 2.09 [0.13, 32.97] | 2018 |
| **Subtotal (95% CI)** | | 2392 | | 8380 | 73.1% | 2.24 [1.71, 2.94] | |
| Total events | 83 | | 141 | | | | |

Heterogeneity: Tau² = 0.00; Chi² = 7.21, df = 9 (P = 0.62); I² = 0%
Test for overall effect: Z = 5.83 (P < 0.00001)

| | | | | | | | |
|---|---|---|---|---|---|---|---|
| **≥75 years old vs. <75 years old** | | | | | | | |
| Cook et al. 2012 | 11 | 151 | 17 | 1292 | 10.2% | 5.54 [2.64, 11.60] | 2012 |
| Booth et al. 2015 | 14 | 186 | 31 | 1124 | 14.8% | 2.73 [1.48, 5.03] | 2015 |
| Nardo et al. 2016 | 1 | 21 | 4 | 128 | 1.2% | 1.52 [0.18, 12.98] | 2016 |
| Gandy et al. 2018 | 1 | 29 | 1 | 158 | 0.7% | 5.45 [0.35, 84.66] | 2018 |
| **Subtotal (95% CI)** | | 387 | | 2702 | 26.9% | 3.54 [2.25, 5.57] | |
| Total events | 27 | | 53 | | | | |

Heterogeneity: Tau² = 0.00; Chi² = 2.81, df = 3 (P = 0.42); I² = 0%
Test for overall effect: Z = 5.45 (P < 0.00001)

| | | | | | | | |
|---|---|---|---|---|---|---|---|
| **Total (95% CI)** | | 2779 | | 11082 | 100.0% | 2.53 [2.00, 3.21] | |
| Total events | 110 | | 194 | | | | |

Heterogeneity: Tau² = 0.00; Chi² = 13.07, df = 13 (P = 0.44); I² = 1%
Test for overall effect: Z = 7.68 (P < 0.00001)
Test for subgroup differences: Chi² = 2.86, df = 1 (P = 0.09), I² = 65.1%

**Fig 3. Forest plots of postoperative outcomes of liver resection for CRLM in older vs. younger patients.** The following outcomes were analyzed: a. overall postoperative complications (n), b. hospital stay (days), and c. mortality (n).

### a. Overall Survival

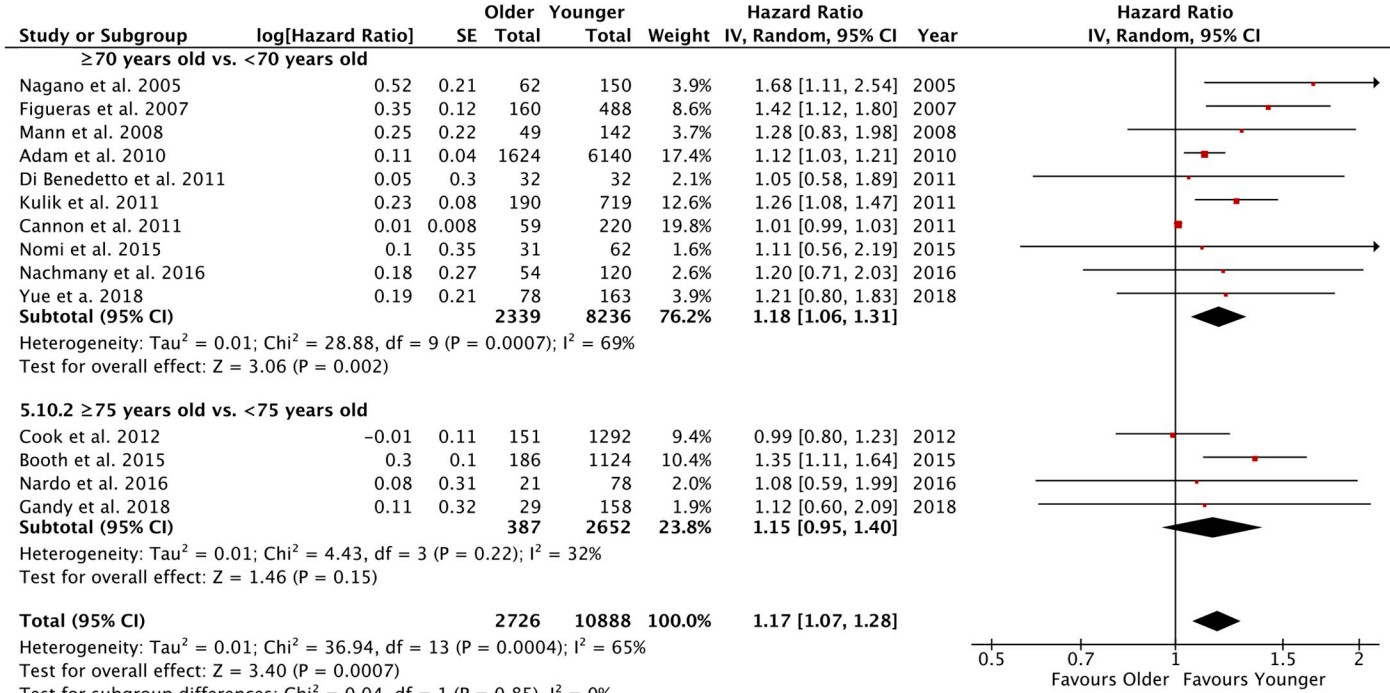

### a. Disease-Free Survival

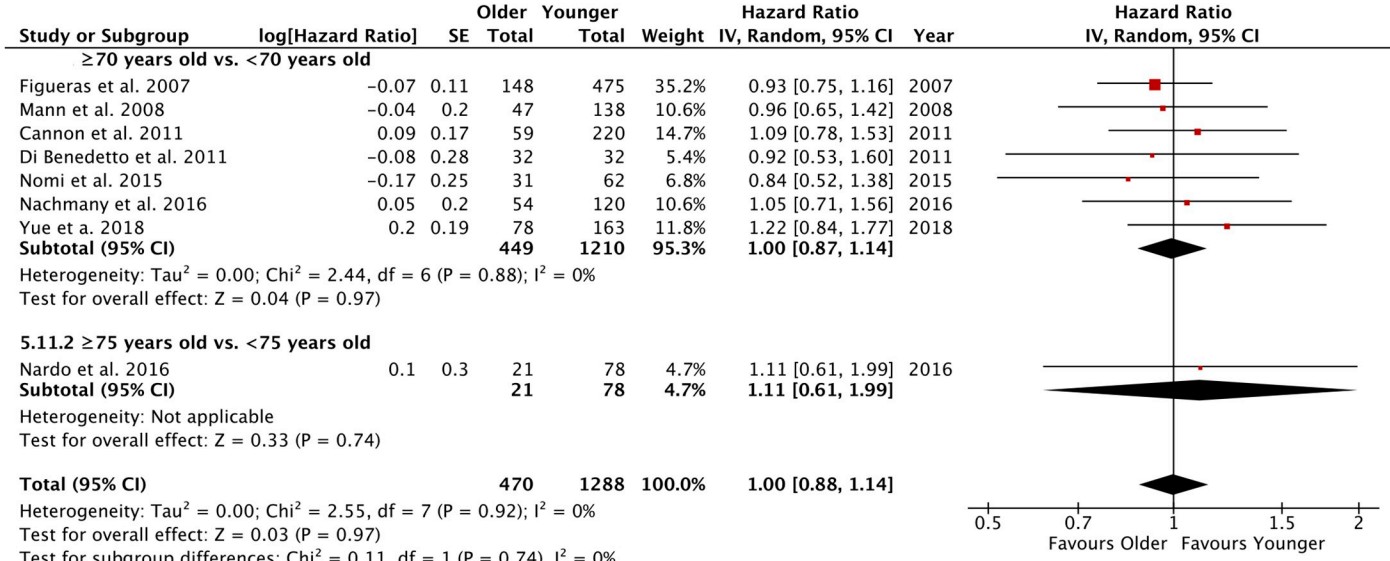

**Fig 4. Forest plots of survival rates of older vs. younger patients after liver resection for CRLM.** The following outcomes were analyzed: a. overall survival and b. disease-free survival.

### a. Blood Loss

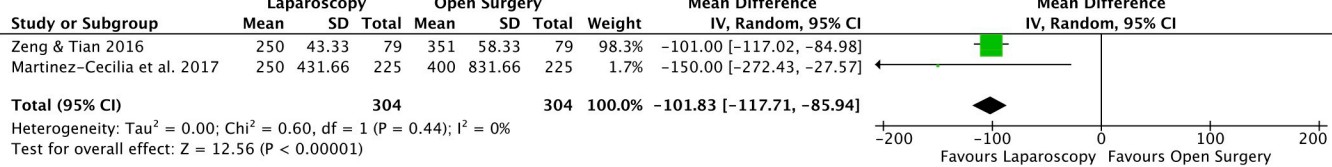

| Study or Subgroup | Laparoscopy Mean | SD | Total | Open Surgery Mean | SD | Total | Weight | Mean Difference IV, Random, 95% CI |
|---|---|---|---|---|---|---|---|---|
| Zeng & Tian 2016 | 250 | 43.33 | 79 | 351 | 58.33 | 79 | 98.3% | −101.00 [−117.02, −84.98] |
| Martinez−Cecilia et al. 2017 | 250 | 431.66 | 225 | 400 | 831.66 | 225 | 1.7% | −150.00 [−272.43, −27.57] |
| **Total (95% CI)** | | | **304** | | | **304** | **100.0%** | **−101.83 [−117.71, −85.94]** |

Heterogeneity: Tau² = 0.00; Chi² = 0.60, df = 1 (P = 0.44); I² = 0%
Test for overall effect: Z = 12.56 (P < 0.00001)

### b. Operative Time

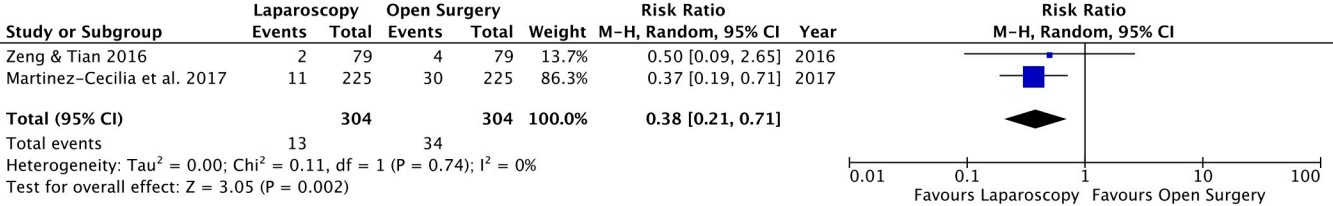

| Study or Subgroup | Laparoscopy Mean | SD | Total | Open Surgery Mean | SD | Total | Weight | Mean Difference IV, Random, 95% CI | Year |
|---|---|---|---|---|---|---|---|---|---|
| Zeng & Tian 2016 | 200 | 13.3 | 79 | 160 | 16.6 | 79 | 53.2% | 40.00 [35.31, 44.69] | 2016 |
| Martinez−Cecilia et al. 2017 | 230 | 87.5 | 225 | 225 | 105 | 225 | 46.8% | 5.00 [−12.86, 22.86] | 2017 |
| **Total (95% CI)** | | | **304** | | | **304** | **100.0%** | **23.60 [−10.63, 57.84]** | |

Heterogeneity: Tau² = 568.12; Chi² = 13.80, df = 1 (P = 0.0002); I² = 93%
Test for overall effect: Z = 1.35 (P = 0.18)

### c. Major Postoperative Complications

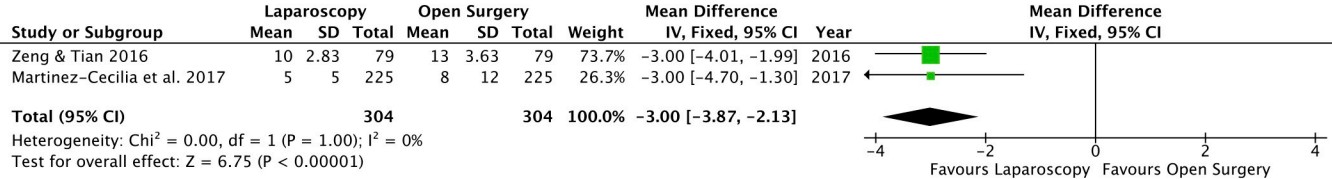

| Study or Subgroup | Laparoscopy Events | Total | Open Surgery Events | Total | Weight | Risk Ratio M−H, Random, 95% CI | Year |
|---|---|---|---|---|---|---|---|
| Zeng & Tian 2016 | 2 | 79 | 4 | 79 | 13.7% | 0.50 [0.09, 2.65] | 2016 |
| Martinez−Cecilia et al. 2017 | 11 | 225 | 30 | 225 | 86.3% | 0.37 [0.19, 0.71] | 2017 |
| **Total (95% CI)** | | **304** | | **304** | **100.0%** | **0.38 [0.21, 0.71]** | |
| Total events | 13 | | 34 | | | | |

Heterogeneity: Tau² = 0.00; Chi² = 0.11, df = 1 (P = 0.74); I² = 0%
Test for overall effect: Z = 3.05 (P = 0.002)

### d. Hospital Stay

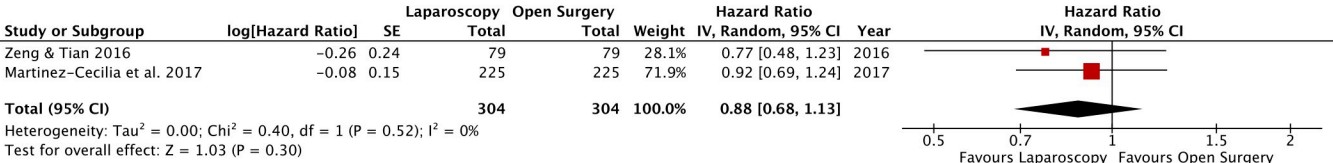

| Study or Subgroup | Laparoscopy Mean | SD | Total | Open Surgery Mean | SD | Total | Weight | Mean Difference IV, Fixed, 95% CI | Year |
|---|---|---|---|---|---|---|---|---|---|
| Zeng & Tian 2016 | 10 | 2.83 | 79 | 13 | 3.63 | 79 | 73.7% | −3.00 [−4.01, −1.99] | 2016 |
| Martinez−Cecilia et al. 2017 | 5 | 5 | 225 | 8 | 12 | 225 | 26.3% | −3.00 [−4.70, −1.30] | 2017 |
| **Total (95% CI)** | | | **304** | | | **304** | **100.0%** | **−3.00 [−3.87, −2.13]** | |

Heterogeneity: Chi² = 0.00, df = 1 (P = 1.00); I² = 0%
Test for overall effect: Z = 6.75 (P < 0.00001)

### e. Overall Survival

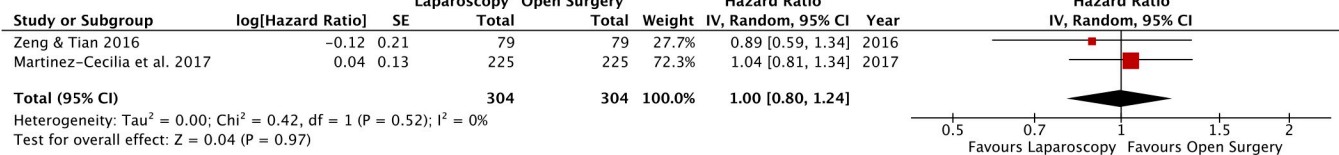

| Study or Subgroup | log[Hazard Ratio] | SE | Laparoscopy Total | Open Surgery Total | Weight | Hazard Ratio IV, Random, 95% CI | Year |
|---|---|---|---|---|---|---|---|
| Zeng & Tian 2016 | −0.26 | 0.24 | 79 | 79 | 28.1% | 0.77 [0.48, 1.23] | 2016 |
| Martinez−Cecilia et al. 2017 | −0.08 | 0.15 | 225 | 225 | 71.9% | 0.92 [0.69, 1.24] | 2017 |
| **Total (95% CI)** | | | **304** | **304** | **100.0%** | **0.88 [0.68, 1.13]** | |

Heterogeneity: Tau² = 0.00; Chi² = 0.40, df = 1 (P = 0.52); I² = 0%
Test for overall effect: Z = 1.03 (P = 0.30)

### f. Disease-free Survival

| Study or Subgroup | log[Hazard Ratio] | SE | Laparoscopy Total | Open Surgery Total | Weight | Hazard Ratio IV, Random, 95% CI | Year |
|---|---|---|---|---|---|---|---|
| Zeng & Tian 2016 | −0.12 | 0.21 | 79 | 79 | 27.7% | 0.89 [0.59, 1.34] | 2016 |
| Martinez−Cecilia et al. 2017 | 0.04 | 0.13 | 225 | 225 | 72.3% | 1.04 [0.81, 1.34] | 2017 |
| **Total (95% CI)** | | | **304** | **304** | **100.0%** | **1.00 [0.80, 1.24]** | |

Heterogeneity: Tau² = 0.00; Chi² = 0.42, df = 1 (P = 0.52); I² = 0%
Test for overall effect: Z = 0.04 (P = 0.97)

**Fig 5. Forest plots of operative and postoperative outcomes of laparoscopic liver resection vs. open liver resection for CRLM in older patients.** The following outcomes were analyzed: a. blood loss (mL); b. operative time (min); c. major postoperative complications (n); d. hospital stay (days); e. overall survival; and f. disease-free survival.

## Discussion

The present systematic review and meta-analysis comprised studies published in the last 19 years that investigated the outcomes of surgical and non-surgical regional treatments for mCRC in older patients. The qualitative synthesis revealed that the current pertinent literature is lacking RCTs while only retrospective studies with heterogonous study design, study populations, and study outcomes can be found. Notwithstanding, the review question remains extremely actual and critical, considering the increasing life expectancy of the general population and the compelling incidence of mCRC in older patients [5–8].

The treatment of mCRC requires a multidisciplinary approach that must include chemotherapy, surgery or other regional strategies to be curative [15, 16]. This results in a complex management burden by morbidity and adverse events that can be more frequent and severe in already frail patients. Indeed, it appears from the literature that although all treatments can be considered for older patients with mCRC, these patients are less likely to receive aggressive curative-intent therapies compared to younger ones [40, 42, 54, 57]. The main reason is found in the anticipated risk of mortality and morbidity that guides clinicians to choose a less aggressive approach. However, overall, the literature states that the patient's advanced chronologic age cannot be considered as an absolute medical contraindication for regional treatments in case of mCRC [49, 58], including surgical resection of CRC metastases [51, 52, 55–57, 62, 64]. Pooled data analyses support this therapeutic attitude, since no difference was found in terms of operative parameters and postoperative complications between older (both aged ≥70 and 75 years) and younger patients. However, older patients are at 2 to 3-fold increased risk of postoperative mortality compared to younger patients. This finding may reflect the increased ASA score and comorbidity index usually observed in older patients undergoing treatments for mCRC and may reflect a lower likelihood of older patients to recover after postoperative complications. Indeed, in most of the studies, a significant imbalance between the two age groups was noted for these clinical variables, but their impact as covariates is hardly assessable. Notwithstanding it, we may hypothesize that although the estimated rate of postoperative complications is similar between older and younger patients, whenever a postoperative complication occurs, this is more likely to be fatal in the older patient group. Conversely, no significant age-related difference was noted for mortality rate after 90Y-RE.

Pooled survival analyses showed that older age is associated with worse survival after surgery for mCRC, as expected. However, disease-free survival rates appear not to be affected by the patients' age. These data suggest that the risk of cancer recurrence is similar between older and younger patients, underlying a negligent impact of chronological age on DFS, and it can be interpreted as indirect proof of a similar success rate of CRLM surgical resection in both older and younger CRC patients.

Considering the surgical approach to liver resection, the two selected comparative studies demonstrated that there is no difference in terms of OS and DFS outcomes between open surgery and laparoscopy in older patients [65, 66]. However, open surgery is associated with a 2.6 times greater risk of major postoperative complications [65, 66] and overall postoperative morbidity compared to laparoscopy [66]. These findings are in accordance with the results of the OSLO-COMET RCT that demonstrated the superiority of laparoscopic liver resection for mCRC [68], and supports the use of laparoscopy also in selected older patients. Moreover, they corroborate the findings of a recent meta-analysis that evaluated the outcomes of all types of laparoscopic liver resection (for both benign and malignant lesions) in older patients, and reported significantly better intra-operative (e.g., bleeding) and postoperative outcomes (e.g., severe Dindo-Clavien complications) for laparoscopy vs. open surgery [69]. It must be noted that the surgical approach chosen may have a relevant impact on postoperative outcomes,

especially in a higher-risk patient population such as older patients. Indeed, minimizing surgical trauma can facilitate the patient's recovery and results in benefits from both the patient's perspective and the healthcare system's perspective. Thus, age should not be regarded as a contraindication for laparoscopy; rather, this approach should be preferred whenever an adequate surgeon's proficiency and experience is insured [70].

The results of the present systematic review and meta-analysis should be interpreted in light of the study limitations, among which we must acknowledge the clinical heterogeneity of the included studies, the lack of a standardized age definition and outcomes, and the lack of variable adjustments on potential confounders such as the score of frailty. It is highly difficult to assess the potential bias linked to the selection of patients for whom the treatment is indicated. Moreover, all included studies have a retrospective design, which increases the risk of selection and reporting bias. However, we attempted to reduce other potential sources of bias by performing a literature search limited to the time period 2000–2019, which avoided important discrepancies that may be found when comparing treatment protocols and indications prior to the year 2000. Finally, the literature search and evaluation were performed by two independent and blind reviewers, a digestive surgeon and a geriatric oncology specialist, who were advised by a team of multidisciplinary contributors including oncologists, geriatrics, gastroenterologists, hepatologists, radiologists, and methodologists.

## Key-points and future research perspectives

- Standardized treatment protocols and international guidelines are eagerly awaited in order to limit unjustified treatment differences based on the patient's age.

- A standardized definition of older patient, comprising a validated evaluation of frailty, should be systematically used in studies focusing on the treatment of mCRC in order to allow a better comparability of study outcomes and provide a stronger evidence-based interpretation of study results.

- Whenever a curative-intent surgical resection for mCRC can be attempted, this should be encouraged and tailored to the patient's performance status, comorbidity index and willingness to receive treatments rather than on age. These factors should be evaluated preoperatively in multidisciplinary meetings, which should involve oncologists, geriatrics, gastroenterologists, hepatologists, radiologists, and liver surgeons, to insure the best patient global management.

- Prospective single center or multicenter registers are required to assess patient-centered outcomes, such as the quality of life for CRC survivors with or without a recurrence of cancer. These outcomes are currently completely disregarded in the literature.

- Development of treatment protocols tailored to older patient populations will consequently have an impact on the economic burden of CRC, with direct and indirect medical costs that are expected to increase due to population changes alone [71]. In the case of mCRC, it is also extremely relevant to investigate the financial burden of cancer for the patients and their family to assist health care policy makers in their efforts to improve the quality of survivorship in older patients.

## Conclusion

The present systematic review and meta-analysis suggest that older patients undergoing surgery and local ablation treatments for mCRC have similar operative outcomes and similar

postoperative complication rates as younger patients. These treatment options should not be disregarded *a priori* in patients aged 65 years or more, but clinicians should be aware that older patients are at an increased risk of postoperative mortality and have a worse overall survival compared to younger patients. These two outcomes may reflect the competitive effects of comorbidity and frailty of the older population, which need to be further evaluated in future studies.

## Supporting information

**S1 Checklist. PRISMA 2009 checklist.**
(DOC)

**S1 Fig. Number of publications per year on the local ablation treatments for colorectal cancer metastases in elderly patients.**
(TIFF)

**S2 Fig. Forest plots of estimated blood loss in older ($\geq$70 years old) vs. younger patients (<70 years old) undergoing liver resection for CRLM.**
(TIFF)

**S3 Fig. Forest plots of postoperative outcomes of liver resection for CRLM in older vs. younger patients.** The following outcomes were analyzed: a. bile leak (n) and b. liver failure (n).
(TIFF)

**S4 Fig. Forest plots of pulmonary complications in older ($\geq$70 years old) vs. younger patients (<70 years old) after liver resection for CRLM.**
(TIFF)

**S5 Fig. Forest plots of major postoperative complications (Dindo-Clavien III or more) after liver resection for CRLM in older vs. younger patients.**
(TIFF)

**S6 Fig. Forest plots of mortality rate in older ($\geq$70 years old) vs. younger patients (<70 years old) after Y90-RE for CRLM.**
(TIFF)

**S1 Table. Study quality assessment using Newcastle-Ottawa scale (NOS).**
(DOCX)

## Acknowledgments

The authors would like to thank the Societé Francophone d'Oncogeriatrie (SoFOG) for their support.

## Author Contributions

**Conceptualization:** Nicola de'Angelis, Capucine Baldini, Raffaele Brustia, Patrick Pessaux, Daniele Sommacale, Alexis Laurent, Vania Tacher, Hicham Kobeiter, Alain Luciani, Elena Paillaud, Thomas Aparicio, Florence Canuï-Poitrine, Evelyne Liuu.

**Data curation:** Nicola de'Angelis, Capucine Baldini, Florence Canuï-Poitrine, Evelyne Liuu.

**Formal analysis:** Nicola de'Angelis.

**Investigation:** Nicola de'Angelis, Vania Tacher, Hicham Kobeiter, Elena Paillaud.

**Methodology:** Nicola de'Angelis, Alexis Laurent, Vania Tacher, Elena Paillaud, Thomas Aparicio, Florence Canuï-Poitrine, Evelyne Liuu.

**Project administration:** Elena Paillaud, Thomas Aparicio, Florence Canuï-Poitrine.

**Resources:** Thomas Aparicio.

**Supervision:** Patrick Pessaux, Alain Luciani, Elena Paillaud, Florence Canuï-Poitrine, Evelyne Liuu.

**Validation:** Capucine Baldini, Raffaele Brustia, Bertrand Le Roy, Alain Luciani, Elena Paillaud, Thomas Aparicio, Evelyne Liuu.

**Visualization:** Bertrand Le Roy, Alain Luciani.

**Writing – original draft:** Nicola de'Angelis, Evelyne Liuu.

**Writing – review & editing:** Nicola de'Angelis, Capucine Baldini, Raffaele Brustia, Patrick Pessaux, Daniele Sommacale, Alexis Laurent, Bertrand Le Roy, Vania Tacher, Hicham Kobeiter, Alain Luciani, Elena Paillaud, Thomas Aparicio, Florence Canuï-Poitrine, Evelyne Liuu.

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
