## [Decision Letter · Decision Letter 0]

23 Dec 2019

PONE-D-19-31837

Regional treatments for colorectal cancer metastases in older patients: A systematic review and meta-analysis

PLOS ONE

Dear Dr. de'Angelis,

Thank you for submitting your manuscript to PLOS ONE. After careful consideration, we feel that it has merit but does not fully meet PLOS ONE’s publication criteria as it currently stands. Therefore, we invite you to submit a revised version of the manuscript that addresses the points raised during the review process.

We would appreciate receiving your revised manuscript by Jan 31 2020 11:59PM. To enhance the reproducibility of your results, we recommend that if applicable you deposit your laboratory protocols in protocols.io, where a protocol can be assigned its own identifier (DOI) such that it can be cited independently in the future. For instructions see: http://journals.plos.org/plosone/s/submission-guidelines#loc-laboratory-protocols

We look forward to receiving your revised manuscript.

Kind regards,

Giuseppe Nigri

Academic Editor

PLOS ONE

2. We noticed you have some minor occurrence(s) of overlapping text with the following previous publication(s), which needs to be addressed:

http://doi.org/10.1001/jamasurg.2016.5665

http://dx.doi.org/10.3748/wjg.v21.i39.11185

In your revision ensure you cite all your sources (including your own works), and quote or rephrase any duplicated text outside the Methods section. Further consideration is dependent on these concerns being addressed.

3. To comply with the items on the PRISMA checklist, please structure the abstract in subheadings.

4. Thank you for stating the following in the Competing Interests section: "The authors have declared that no competing interests exist."

We note that one or more of the authors are employed by a commercial company: "Pr P Pessaux is orator for Integra and co-founder of VirtualiSurg."

5.  One of the noted authors is a group or consortium: SoFOG (Societé Francophone d’Oncogeriatrie)

In addition to naming the author group, please list the individual authors and affiliations within this group in the acknowledgments section of your manuscript. Please also indicate clearly a lead author for this group along with a contact email address.

Reviewers' comments:

Reviewer's Responses to Questions

**Comments to the Author**

1. Is the manuscript technically sound, and do the data support the conclusions?

Reviewer #1: Partly

Reviewer #2: Yes

2. Has the statistical analysis been performed appropriately and rigorously? 

Reviewer #1: Yes

Reviewer #2: Yes

3. Have the authors made all data underlying the findings in their manuscript fully available?

Reviewer #1: Yes

Reviewer #2: Yes

4. Is the manuscript presented in an intelligible fashion and written in standard English?

Reviewer #1: Yes

Reviewer #2: Yes

5. Review Comments to the Author

Reviewer #1: In this manuscript the authors focus on an interesting clinical question concerning the management of elder patients affected by CRLM. The article is well written, the PRISMA guidelines are well followed, the statistics are good and the conclusions are fairly supported by the results. I have some minor comments that should be addressed by the authors.

1) I feel that there is some discrepancy and confusion with terminology throughout the manuscript: with "regional treatments" i was expecting only non-surgical treatments while you include also liver resections. I would clearly separate the two and be more appealing even in your title as it is my opinion that you are giving us some sound conclusions that liver resections are a good options in older patients. This is the main message of you article rather than regional and non-surgical treatments.

2) There are too many tables and these are over written. It is good to me because i had the chance to see that you carefully screened the manuscripts. However, this is too much even in the setting of editorial policies i guess. I would merge some tables and cut down the number of words in each field. Give results, not comments.

3) Similarly, the number of figures is way too much. Figure 2,3,4, and 5 for example, i would combine and give one figure with the Forest plots of intraoperative outcomes and one with the postoperative outcomes. We don't need specific results for bile leak and ascites.

Reviewer #2: Authors correctly stated that the treatment of older patients is a hot topic in all the fields of medicine. The treatment of liver metastases from colorectal cancer fell in this topic.

A great effort has been made by them in the evaluation of paper dealing with this argument and published in the English written literature. A total of 29 paper has been selected and evaluated with a rigorous methodology. The paper support the performance of curative treatments even in the setting of older patients, since the results published in the literature and analyzed in this meta-analysis do not show inferior results when compared with those achieved in younger patients.

Included studies are retrospective.

It is obvious that older patients are more prone to develop postoperative complications and this came out also from their analysis.

Few considerations should be evaluated and possibly included in the text.

1) There is no mention of the stage of the diseases that has been treated. It is possible that older patients have been treated for less invasive in term of number of metastatic nodules and diameter of the tumor. At page 8 (Data extraction and quality assessment) there is no mention of the stage of the disease that have been operated. Probably this could not be retrieved from papers. Nevertheless, I believe it should be mention as not retrievable and included in the discussion chapter.

2) At page 8 it is stated that the “type of intervention” has been included into the considered variable. When dealing with surgical treatments, “type of intervention” usually refers to the extension of the hepatectomy performed. Maybe the Authors should better clarify the definition.

3) Could it be possible to retrieve the extension of the hepatectomy performed from the collected manuscripts? It is possible that major hepatectomies are rarely permed in older patients. If this data could not be collected, it should be stated in the “data extraction” subchapter and commented in the “Discussion” chapter.

4) Page 50 – Key-point #3. Up to today, the only “curative-intent” strategy in the treatment of colorectal liver metastasis is the surgical resection. It should be stated clearly at this point, not to leading to misunderstandable messages to the readers.

5) As a consequence, it is advisable to include among the key-points, that these patients should be evaluated for treatment by a multidisciplinary committee that should mandatory include the figure of a hepatic surgeon.

6. PLOS authors have the option to publish the peer review history of their article (what does this mean?). If published, this will include your full peer review and any attached files.

Reviewer #1: No

Reviewer #2: Yes: Gian Luca Grazi, MD

---

## [Author Response · Author response to Decision Letter 0]

30 Jan 2020

Response to Reviewers – Manuscript PONE-D-19-31837

A: We revised the manuscript for conformity to style requirements. 

2. We noticed you have some minor occurrence(s) of overlapping text with the following previous publication(s), which needs to be addressed:

http://doi.org/10.1001/jamasurg.2016.5665

http://dx.doi.org/10.3748/wjg.v21.i39.11185

In your revision ensure you cite all your sources (including your own works), and quote or rephrase any duplicated text outside the Methods section. Further consideration is dependent on these concerns being addressed.

A: We used iThenticate software to detect all overlaps with previous publications and thus we provided citations or rephrasing whenever necessary allover the manuscript, in particularly for the two articles that you indicated. The matching rate did not exceed 1% for all publications cited in the text. 

3. To comply with the items on the PRISMA checklist, please structure the abstract in subheadings.

A: We structured the abstract in subheadings.

4. Thank you for stating the following in the Competing Interests section: "The authors have declared that no competing interests exist."

A: We completed this section.

We note that one or more of the authors are employed by a commercial company: "Pr P Pessaux is orator for Integra and co-founder of VirtualiSurg."

A: We updated the Funding section with more details. Please note that we did not receive any external financial support or research grants to perform this systematic review and meta-analysis. The institutional support is limited to the authors’ own salary. Pr Pessaux declared his relationship with a commercial affiliation, which however has NO role in the present study. No competing interest is declared in relation with the matter of the present study. 

A: We updated the competing interest statement. 

A: Done

5. One of the noted authors is a group or consortium: SoFOG (Societé Francophone d’Oncogeriatrie)

In addition to naming the author group, please list the individual authors and affiliations within this group in the acknowledgments section of your manuscript. Please also indicate clearly a lead author for this group along with a contact email address.

A: We did not intend the scientific society (SoFOG) as author group of the manuscript. Two authors (EP, TA) are presidents of this scientific society who promoted the work (only scientifically, no financial support, no role in study design, data interpretation or publication policy). We thus kept the SoFOG only in the acknowledgments section. 

Reviewers’ comments 

Reviewers’ comments are indicated with R, authors’ responses with A.

Reviewer #1

R: In this manuscript the authors focus on an interesting clinical question concerning the management of elder patients affected by CRLM. The article is well written, the PRISMA guidelines are well followed, the statistics are good and the conclusions are fairly supported by the results. 

A: Thank you for your positive comment and for the time spent revising our manuscript. 

I have some minor comments that should be addressed by the authors.

R1) I feel that there is some discrepancy and confusion with terminology throughout the manuscript: with "regional treatments" i was expecting only non-surgical treatments while you include also liver resections. I would clearly separate the two and be more appealing even in your title as it is my opinion that you are giving us some sound conclusions that liver resections are a good options in older patients. This is the main message of you article rather than regional and non-surgical treatments.

A: We agree with the reviewer, and we opted for a more precise title rather than a generic one that can be potentially confusing. Consequently, we revised the entire manuscript for consistency. 

2) There are too many tables and these are over written. It is good to me because i had the chance to see that you carefully screened the manuscripts. However, this is too much even in the setting of editorial policies i guess. I would merge some tables and cut down the number of words in each field. Give results, not comments.

A: We agree with the reviewer. We merged the 6 tables into 3 tables and we did our best to reduce the number of word and summarized the content of the tables in a format that will be easer for the reader. 

3) Similarly, the number of figures is way too much. Figure 2,3,4, and 5 for example, i would combine and give one figure with the Forest plots of intraoperative outcomes and one with the postoperative outcomes. We don't need specific results for bile leak and ascites.

A: We agree with the reviewer. As suggested, we merged several forest plots to have only 5 figures, the rest is proposed as supplemental materials. 

Reviewer #2

Authors correctly stated that the treatment of older patients is a hot topic in all the fields of medicine. The treatment of liver metastases from colorectal cancer fell in this topic.

A great effort has been made by them in the evaluation of paper dealing with this argument and published in the English written literature. A total of 29 paper has been selected and evaluated with a rigorous methodology. The paper support the performance of curative treatments even in the setting of older patients, since the results published in the literature and analyzed in this meta-analysis do not show inferior results when compared with those achieved in younger patients.

Included studies are retrospective.

It is obvious that older patients are more prone to develop postoperative complications and this came out also from their analysis.

A: Thank you for your positive comments and for the time spent revising our manuscript.

Few considerations should be evaluated and possibly included in the text.

R1) There is no mention of the stage of the diseases that has been treated. It is possible that older patients have been treated for less invasive in term of number of metastatic nodules and diameter of the tumor. At page 8 (Data extraction and quality assessment) there is no mention of the stage of the disease that have been operated. Probably this could not be retrieved from papers. Nevertheless, I believe it should be mention as not retrievable and included in the discussion chapter.

A: We agree with the reviewer that it is important to report the stage of disease and metastasis characteristics and whether these factors were significantly different between older and younger patients receiving liver surgery. In Table 1 we reported the detailed characteristics of metastases for each articles included and we indicated whenever there is any significant group difference. We had also reported the presence of imbalance between groups (and on which variables). However, in this revised version, as suggested by reviewer #1, we simplified the tables to cut down the number at 3 tables only. We had to remove the column about group imbalances. However, we checked and no mention of disease stage was ever made by the different authors. This is hardly assessable from the selected paper, but we mentioned it in the results section. 

R2) At page 8 it is stated that the “type of intervention” has been included into the considered variable. When dealing with surgical treatments, “type of intervention” usually refers to the extension of the hepatectomy performed. Maybe the Authors should better clarify the definition.

A: Thank you, we better clarified it in the text. 

R3) Could it be possible to retrieve the extension of the hepatectomy performed from the collected manuscripts? It is possible that major hepatectomies are rarely permed in older patients. If this data could not be collected, it should be stated in the “data extraction” subchapter and commented in the “Discussion” chapter.

A: Whenever available in the selected studies, these details are presented in Table 1 (column Type of intervention), specifying if major or minor hepatectomies and if differences between the groups. 

R4) Page 50 – Key-point #3. Up to today, the only “curative-intent” strategy in the treatment of colorectal liver metastasis is the surgical resection. It should be stated clearly at this point, not to leading to misunderstandable messages to the readers.

A: We agree with the reviewer and we revised the key point accordingly. 

R5) As a consequence, it is advisable to include among the key-points, that these patients should be evaluated for treatment by a multidisciplinary committee that should mandatory include the figure of a hepatic surgeon.

A: We agree with the reviewer and we revised the key point accordingly.

---

## [Editor Report · Decision Letter 1]

12 Mar 2020

Surgical and regional treatments for colorectal cancer metastases in older patients: A systematic review and meta-analysis

PONE-D-19-31837R1

Dear Dr. de'Angelis,

We are pleased to inform you that your manuscript has been judged scientifically suitable for publication and will be formally accepted for publication once it complies with all outstanding technical requirements.

With kind regards,

Giuseppe Nigri

Academic Editor

PLOS ONE

---

## [Editor Report · Acceptance letter]

23 Mar 2020

PONE-D-19-31837R1 

Surgical and regional treatments for colorectal cancer metastases in older patients: A systematic review and meta-analysis 

Dear Dr. de'Angelis:

I am pleased to inform you that your manuscript has been deemed suitable for publication in PLOS ONE. Congratulations! Your manuscript is now with our production department. 

With kind regards,

on behalf of

Dr. Giuseppe Nigri 

Academic Editor

PLOS ONE